# SHP-2 and PD-1-SHP-2 signaling regulate myeloid cell differentiation and antitumor responses

Anthos Christofides[1,2,3,10,14], Xanthi-Lida Katopodi[4,5,6,14], Carol Cao[1,2,7], Dimitra Karagkouni[4,5,6], Konstantinos Aliazis [1,2,3], Sasitorn Yenyuwadee[1,2,3,11], Halil-Ibrahim Aksoylar[1,2,3], Rinku Pal[1,2,3], Mohamed A. A. Mahmoud[1,2,12], Laura Strauss[1,2,3,13], Natalia M. Tijaro-Ovalle[1,2,10], Louis Boon[8], John Asara[3], Ioannis S. Vlachos[4,5,6,9], Nikolaos Patsoukis[1,2,3] & Vassiliki A. Boussiotis [1,2,3] ✉

The inhibitory receptor PD-1 suppresses T cell activation by recruiting the phosphatase SHP-2. However, mice with a T-cell-specific deletion of SHP-2 do not have improved antitumor immunity. Here we showed that mice with conditional targeting of SHP-2 in myeloid cells, but not in T cells, had diminished tumor growth. RNA sequencing (RNA-seq) followed by gene set enrichment analysis indicated the presence of polymorphonuclear myeloid-derived suppressor cells and tumor-associated macrophages (TAMs) with enriched gene expression profiles of enhanced differentiation, activation and expression of immunostimulatory molecules. In mice with conditional targeting of PD-1 in myeloid cells, which also displayed diminished tumor growth, TAMs had gene expression profiles enriched for myeloid differentiation, activation and leukocyte-mediated immunity displaying >50% overlap with enriched profiles of SHP-2-deficient TAMs. In bone marrow, GM-CSF induced the phosphorylation of PD-1 and recruitment of PD-1-SHP-2 to the GM-CSF receptor. Deletion of SHP-2 or PD-1 enhanced GM-CSF-mediated phosphorylation of the transcription factors HOXA10 and IRF8, which regulate myeloid differentiation and monocytic-moDC lineage commitment, respectively. Thus, SHP-2 and PD-1-SHP-2 signaling restrained myelocyte differentiation resulting in a myeloid landscape that suppressed antitumor immunity.

To escape immunosurveillance, cancer cells have developed mechanisms that mask their immunogenic features, such as expression of ligands for inhibitory receptors that directly inhibit T cell responses by engaging immune inhibitory receptors, such as PD-1 (ref. 1). Tumors alter myeloid cells, which constitute a considerable cellular fraction of the microenvironment, to suppress antitumor responses. However, tumor-infiltrating myeloid cells may also contain immunostimulatory subsets, such as tumor-associated macrophages (TAMs) that produce proinflammatory cytokines and type 1 classical dendritic cells. Conversely, immunosuppressive myeloid cells include protumorigenic TAMs[2] and immature myeloid-derived suppressor cells (MDSCs)[3].

The inhibitory receptor PD-1 blocks T cell activation through a process attributed to the recruitment of the phosphatase SHP-2 to its cytoplasmic tail[4]. Because of this, it was expected that deletion of SHP-2 would abrogate the inhibitory pathway activated downstream of PD-1 receptor. However, T cell-specific deletion of SHP-2 did not improve

antitumor immunity and did not alter antitumor responses of these mice to PD-1 antibody treatment[5] but instead had a detrimental effect on tumor progression[6]. PD-1 is also expressed in common myeloid progenitors (CMPs) and granulocyte/macrophage progenitors (GMPs), which accumulate during cancer-driven emergency myelopoiesis and give rise to immunosuppressive MDSC and TAMs[7]. In tumor-bearing mice with myeloid-specific deletion of PD-1, diminished accumulation of MDSCs was observed in the spleen and tumors, while the output of differentiated effector myeloid cells with monocytic lineage dominance was increased[7]. The molecular mechanisms behind these observations remain unclear.

Temporal activation of SHP-2 is critical for myeloid cell fate. Gain of function mutations in SHP-2 with constitutive phosphatase activation prevent myeloid differentiation and lead to the accumulation of immature myelocytes and development of leukemia[8]. The transcription factors HOXA, which regulate hematopoiesis, are SHP-2 targets[9]. HOXA genes are maximally expressed in committed myeloid progenitors and their dysregulation is associated with leukemia[10]. Tyrosine phosphorylation of the HOXA10 homeodomain during growth factor-induced myelopoiesis decreases its binding affinity for target gene promoters and abrogates HOXA10-induced transcriptional repression, which allows differentiation[11]. SHP-2 also regulates the phosphorylation of the transcription factor IRF8, which is essential for the development of monocytes and DCs from monocyte-DC progenitors (MDPs), while inhibiting neutrophil differentiation[12]. IRF8 phosphorylation at Tyr95 in the conserved IRF domain is mandatory for nuclear translocation and function. SHP-2 dephosphorylates IRF8 and prevents its nuclear localization[13]. Loss of functional IRF8 leads to the generation of MDSCs[14].

Here we showed that in bone marrow myelocytes, GM-CSF induced PD-1 phosphorylation, interaction with SHP-2 and PD-1-SHP-2 recruitment to the GM-CSF receptor. Conditional ablation of either SHP-2 or PD-1 in myeloid cells resulted in augmented phosphorylation of HOXA10 and IRF8 in vitro and enhanced differentiation, activation, phagocytosis and inflammatory responses in myeloid cells of tumor-bearing mice, leading to diminished tumor growth.

## Results

### Myeloid-specific SHP-2 targeting suppresses tumor growth

To dissect the role of PD-1-SHP-2 in antitumor immune responses, we crossed *Shp2^f/f* mice with mice expressing Cre recombinase under the control of the lysozyme (LysM) promoter to induce selective depletion of *Ptpn11* in myeloid cells (*Shp2^f/f^LysM^Cre*) or under the control of the distal Lck promoter to induce selective depletion of *Ptpn11* in mature T cells (*Shp2^f/f^Lck^Cre*). We monitored tumor growth longitudinally for 2 weeks starting on day 7 post subcutaneous implantation of B16-F10 melanoma cells. *Shp2^f/f^LysM^Cre* mice had substantially reduced tumor growth compared to *Shp2^f/f* mice, while tumor growth in *Shp2^f/f^Lck^Cre* was similar to *Shp2^f/f* mice (Fig. 1a). Based on the expression of CD44, CD8⁺ T cells in tumor-draining lymph nodes (dLN) of *Shp2^f/f^LysM^Cre* mice had a more activated state compared to their counterparts in *Shp2^f/f^Lck^Cre* and control *Shp2^f/f* mice (Fig. 1b,c; gating strategy, Supplementary Fig. 1). CD44^hi^CD62L^lo^ T effector cells (hereafter T_EF cells) and CD44⁺CD62L^hi^ central memory T-like cells (T_CM-like cells) from tumor-bearing *Shp2^f/f^LysM^Cre* mice had increased expression of IFN-γ (Fig. 1d), indicating a state of activation and effector function. In contrast, based on the expression of CD44 and IFN-γ, no difference was observed in the activation of dLN CD8⁺ T cells from *Shp2^f/f^Lck^Cre* mice compared to *Shp2^f/f* tumor-bearing mice (Fig. 1b–d).

MDSCs isolated on day 14–16 post-implantation from *Shp2^f/f^LysM^Cre* mice bearing B16-F10 tumors had substantially diminished immunosuppressive capacity compared to *Shp2^f/f^Lck^Cre* and *Shp2^f/f* MDSCs (Fig. 1e), and lower expression of CD38 (Fig. 1f–h), an indicator of immunosuppressive MDSC[15]. PD-1 is expressed in myeloid progenitors and immature myeloid cells in tumor-bearing mice and its ablation switch the fate of myeloid cells toward inflammatory monocytes and

DC[7]. To examine whether PD-1-SHP-2 signaling operated in myeloid cells, tumor-bearing mice were treated with PD-1 antibody on days 9, 11 and 13 after tumor inoculation. *Shp2^f/f* mice displayed a considerable reduction of tumor growth compared to mice receiving control IgG2a (Fig. 2a). In contrast, *Shp2^f/f^LysM^Cre* mice, which had diminished tumor growth compared to *Shp2^f/f* mice, did not substantially benefit from treatment with PD-1 antibody compared to IgG2a (Fig. 2b,c). In addition, the numbers and expression of CD38⁺ MDSC were diminished by PD-1 antibody compared to IgG2a treatment in tumor-bearing *Shp2^f/f* mice, but not in *Shp2^f/f^LysM^Cre* mice (Fig. 2d–f). *Shp2^f/f^LysM^Cre* tumor-bearing mice also exhibited enhanced T cell activation (Fig. 2g,h) and recruitment of CD4⁺ and CD8⁺ T_EF cells in dLN (Fig. 2i,j). Treatment with PD-1 antibody increased the activation of CD8⁺ T cells (Fig. 2g,h) and the numbers of CD4⁺ and CD8⁺ T_EF cells in *Shp2^f/f*, but not in *Shp2^f/f^LysM^Cre* tumor-bearing mice (Fig. 2i,j), which had more T_EF cells in dLN in both IgG2a and PD-1 Ab treatment groups (Fig. 2i). As such, myeloid-specific SHP-2 deletion induced potent antitumor immunity, which was only marginally improved by PD-1 blocking immunotherapy.

### Myeloid-specific SHP-2 deficiency alters the lineage fate of MDSCs

To study the effects of SHP-2 ablation in myeloid cells in more detail, we used an MC17-51 fibrosarcoma mice tumor model, which induces robust cancer-mediated emergency myelopoiesis, leading to considerable output of bone marrow-derived MDSCs and TAMs[16]. On week 2 post subcutaneous tumor inoculation, tumor growth in *Shp2^f/f^LysM^Cre* mice, but not in *Shp2^f/f^Lck^Cre* mice (Fig. 3a,b and Extended Data Fig. 1a,b), was substantially diminished compared to *Shp2^f/f* mice. MDSCs in the mice consist of two major subsets, CD11b⁺Ly6C^hi^Ly6G⁻ monocytic (M-MDSC) and CD11b⁺Ly6C^lo^Ly6G⁺ polymorphonuclear (PMN-MDSC) cells, which have similar morphology and phenotype to normal monocytes and neutrophils, respectively, but distinct functions[3]. We did not observe quantitative differences in tumor-infiltrating myeloid cells (Fig. 3c), but *Shp2^f/f^LysM^Cre* mice had an increased fraction of M-MDSC in tumors (Fig. 3d,e; gating strategy, Supplementary Fig. 2) and an increased ratio of M-MDSC/PMN-MDSC (Fig. 3f) compared to control *Shp2^f/f* tumor-bearing mice. A similar increase of M-MDSC tumor-infiltrating myeloid cells was observed in B16-F10 tumor-bearing *Shp2^f/f^LysM^Cre* mice, which also developed smaller tumors (Extended Data Fig. 1c–g) compared to *Shp2^f/f* tumor-bearing mice. There was an increase in number of CD4⁺ and CD8⁺ T_EF cells (Fig. 3g,h) and an increase in the fraction activated CD44⁺CD8⁺ T cells (Fig. 3i,j) in dLN, a systemic increase of CD4⁺ and CD8⁺ T_EF and T_CM-like cells (Fig. 3k,l) and enhanced T cell activation (Fig. 3m,n) in the spleen of *Shp2^f/f^LysM^Cre* compared to *Shp2^f/f* MC17-51 tumor-bearing mice. No differences were noted in the expression of checkpoint receptors including PD-1, PD-L1, CTLA-4, TIGIT or ICOS in CD4⁺ and CD8⁺ T cells between tumor-bearing *Shp2^f/f* and *Shp2^f/f^LysM^Cre* (Supplementary Fig. 3) or in the numbers and activation of T_reg cell in dLNs (Supplementary Fig. 4a,b). Thus, myeloid-specific SHP-2 ablation led to increased tumor infiltration by Ly6C^hi^ monocytes and concomitant recruitment and activation of T_EF and T_CM cells.

### SHP-2 deficiency induces MDSC differentiation and activation

At day 15 post MC17-51 tumor injection, PMN-MDSC from *Shp2^f/f^LysM^Cre* mice had diminished immunosuppressive function (Fig. 4a) and lower expression of CD38 (Fig. 4b,c). M-MDSC from *Shp2^f/f^LysM^Cre* mice also had substantially lower immunosuppressive capacity (Supplementary Fig. 4c) compared to the respective MDSCs from *Shp2^f/f* mice. M-MDSC in tumors from *Shp2^f/f^LysM^Cre* mice had higher expression of MHC II, CD86 and IFN-γ than their counterparts in *Shp2^f/f* mice (Fig. 4d–g), consistent with an activated and proinflammatory phenotype with improved antigen presentation and costimulation capacity in M-MDSC. Tumor PMN-MDSC in *Shp2^f/f^LysM^Cre* mice had higher expression of IFN-γ than their counterparts in *Shp2^f/f* mice (Fig. 4f,g), indicating that, in the context of cancer, deletion of SHP-2 switched the differentiation of

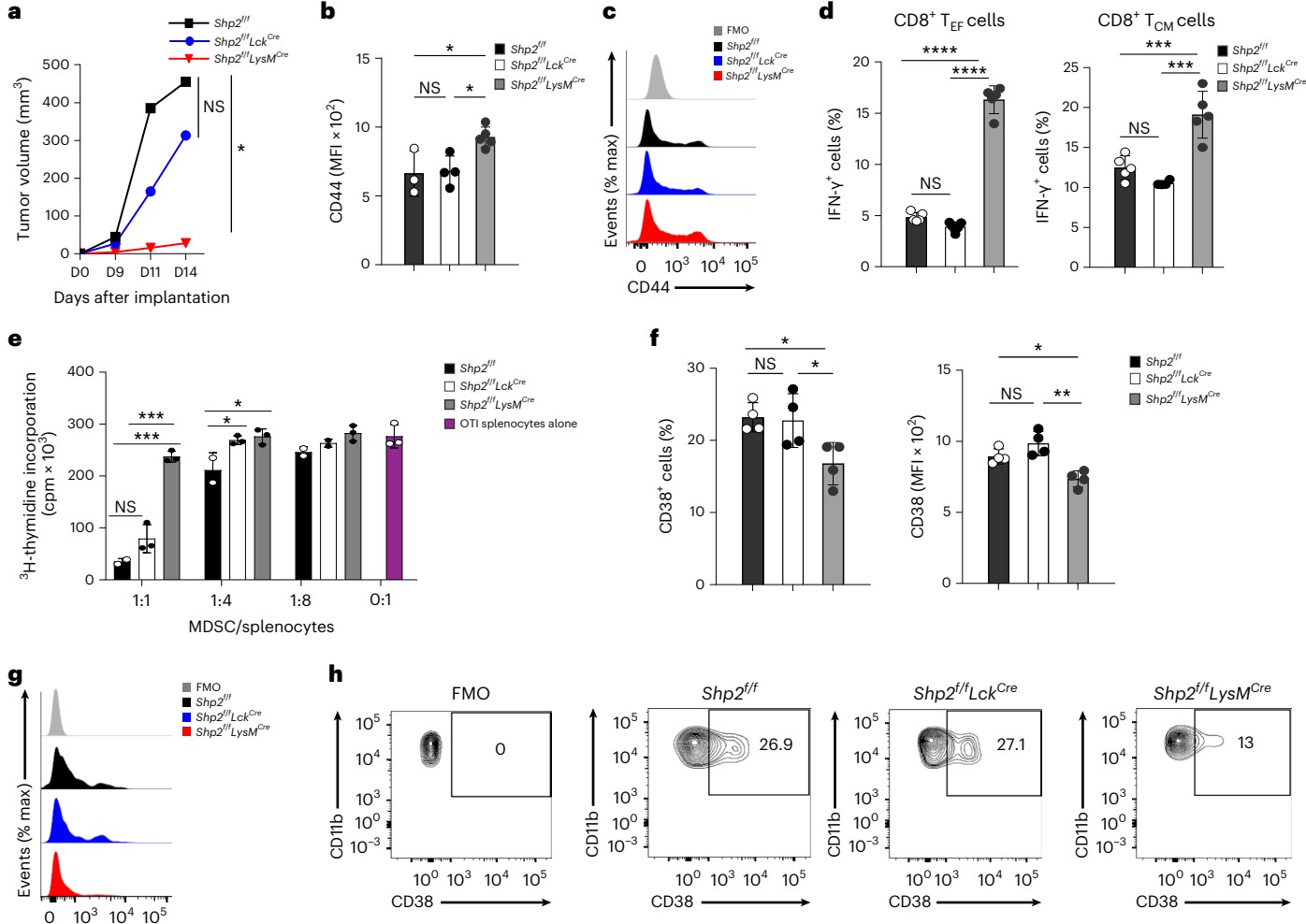

**Fig. 1 | Myeloid-specific SHP-2 deletion diminished tumor progression and suppressive function of MDSC. a**, Tumor volume in *Shp2^{f/f}*, *Shp2^{f/f}Lck^{Cre}* and *Shp2^{f/f}LysM^{Cre}* mice inoculated with B16-F10 melanoma cells ($1 \times 10^5$ cells per mouse) and monitored longitudinally on each day of assessment. Data shown are means of *n* = 6 mice per group and are from one of three independent experiments with reproducible results. **b**–**c**, Quantification (**b**) and representative flow cytometry (**c**) of expression of CD44 in CD8+ T cells isolated from dLNs of mice as in **a**. **d**, Expression of IFN-γ assessed in CD8+ T$_{EM}$ and T$_{CM}$ cells from dLN in mice as in **a**. Mean fluorescence intensity (MFI) ± s.d. results are shown. Results are representative of four independent experiments with *n* = 8 mice per group. **e**, Thymidine incorporation in OTI splenocytes ($2 \times 10^5$

cells per well) stimulated with OVA$_{257–264}$ after the addition of graded numbers of GR1+ MDSCs cells isolated from the spleens of tumor-bearing *Shp2^{f/f}*, *Shp2^{f/f}Lck^{Cre}* and *Shp2^{f/f}LysM^{Cre}* mice. Mean ± s.d. of cpm values are shown. Results are representative of three separate experiments using *n* = 9 mice per group and four technical replicates per condition. **f**,**g**, Representative flow cytometry histograms (**g**) and contour plots (**h**) of CD38 expression on splenic MDSC from *Shp2^{f/f}*, *Shp2^{f/f}Lck^{Cre}* and *Shp2^{f/f}LysM^{Cre}* tumor-bearing mice. Mean percentage ± s.d., MFI ± s.d.% positive cells results are representative of two independent experiments with *n* = 4 and *n* = 6 mice per group and reproducible results. *\*P* = 0.0023–0.0465, *\*\*\*P* = 0.0001–0.0007, *\*\*\*\*P* < 0.0001, ANOVA.

myeloid cells toward proinflammatory neutrophils, and monocytes with enhanced antigen presentation and T cell costimulation capacity.

Upon tumor entry, M-MDSC converts into TAMs that promote tumor progression[3]. To determine the molecular features of the myeloid compartment in tumor-bearing mice, we isolated PMN-MDSC from spleens and TAMs from tumors of *Shp2^{f/f}* and *Shp2^{f/f}LysM^{Cre}* mice on day 15 post MC17-51 tumor implantation and analyzed their gene expression profile by RNA sequencing (RNA-seq). Gene expression analysis showed that among the 17,746 expressed genes, a total of 1,240 genes (842 upregulated and 286 downregulated) were differentially expressed between *Shp2^{f/f}* and *Shp2^{f/f}LysM^{Cre}* MDSC (Fig. 4h,i and Supplementary Table 1). Genes involved in leukocyte differentiation and function[17] were among the top differentially expressed in *Shp2^{f/f}LysM^{Cre}* MDSCs compared to *Shp2^{f/f}* MDSC (Fig. 4i). These included multilineage hematopoietic progenitor genes such as *Klf1*, *Gata2* and *Egr1* and the neutrophil transcription factors *Cebpe* and *Gfi1* (Fig. 4i,j and

Supplementary Table 1), consistent with the GMP origin of PMN-MDSC[3]. A number of granule genes typically expressed by mature neutrophils, in particular, the primary granule proteases neutrophil elastase (*Elane*), proteinase 3 (*Prtn3*), cathepsin G (*Ctsg*) and myeloperoxidase (*Mpo*)[17] (Fig. 4i,j), and genes expressed in mature, fully differentiated granulocytes, such as *S100a8* and *Camp*[17] (Fig. 4i and Supplementary Table 1) were highly expressed in PMN-MDSC from tumor-bearing *Shp2^{f/f}LysM^{Cre}* compared to *Shp2^{f/f}* mice.

Over-representation and gene set enrichment analysis (GSEA) indicated that relative to those isolated from *Shp2^{f/f}* tumor-bearing mice, gene expression profiles of myeloid cells from *Shp2^{f/f}LysM^{Cre}* tumor-bearing mice were enriched for processes involved in the regulation of PRC2-EZH2 targets, phagosome formation, myeloid cell differentiation, neutrophil-mediated immunity, Notch signaling and HOXA targets, and were dominated by functions of antigen processing and presentation, myeloid cell migration, mitosis and autophagy (Fig. 4k,l

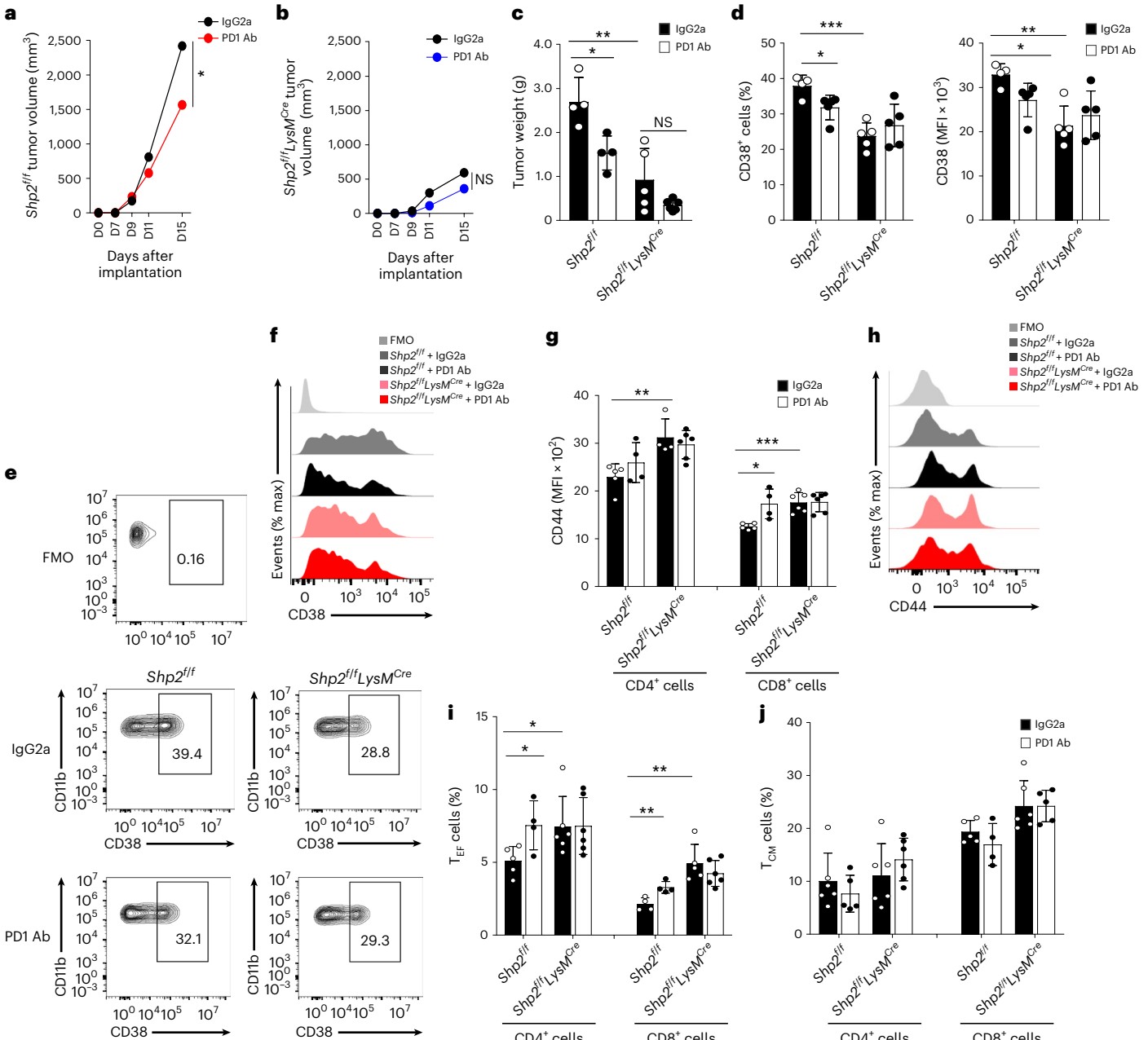

**Fig. 2 | PD-1 blockade induced antitumor responses in *Shp2f/f* mice but not in *Shp2f/fLysMCre* mice. a–c**, Tumor size in *Shp2^{f/f}* (**a**) and *Shp2^{f/f}LysM^{Cre}* mice (**b**) inoculated with B16-F10 melanoma (3 × 10^5 cells per mouse) and treated with PD-1 blocking antibody or IgG2a control on days 9, 11 and 13 after tumor inoculation and measured starting on day 7 (**a**, **b**) or day 15 (**c**). Results show means + s.d. are representative of three experiments *n* = 4 mice per group and two experiments with *n* = 5 mice per group. **d–f**, Quantification (**d**) and representative flow cytometry (**e**, **f**) of CD38 expression in spleen PMN-MDSC isolated from mice as in **a**. **g**,**h**, Quantification (**g**) and representative histograms (**h**) of CD44 expression in CD4^+ and CD8^+ T cells of dLN. **i**,**j**, Frequency of CD4^+ and CD8^+ T_EF cells (**i**) and CD4^+ and CD8^+ T_CM-like cells (**j**) in dLN on *Shp2^{f/f}* versus *Shp2^{f/f}LysM^{Cre}* mice treated with IgG2a or PD-1 Ab mean percentage ± s.d. are shown. Results are from one of three separate experiments with five mice per group. *\*P* = 0.017–0.028, *\*\*P* = 0.0047–0.0075, *\*\*\*P* = 0.0006, unpaired two-sided *t* test.

and Extended Data Fig. 2a). Consistent with these, transcripts of the IFN I-inducible genes *Rsad2*, *Ifit2* and *Cmpk2*, which is required for NLRP3 inflammasome activation downstream of IFNR1 signaling[18], were substantially elevated (Fig. 4i and Supplementary Table 1), while the expression of *Trem2*, a myeloid receptor that transmits intracellular signals promoting immunosuppression function in myeloid cells[19], was substantially decreased (Fig. 4i,j and Supplementary Table 1) in *Shp2^{f/f}LysM^{Cre}* MDSC compared to *Shp2*^{f/f} MDSC. *Shp2^{f/f}LysM^{Cre}* MDSCs also exhibited lower expression of *Msr1*, which promotes neutrophil netosis, a protumorigenic process, and *Xbp1*, a classical marker of the unfolded protein response, which polarizes tumor-infiltrating myeloid cells to

highly immunosuppressive MDSC[20] (Fig. 4i,j and Supplementary Table 1). Thus, myeloid cells in tumor-bearing *Shp2^{f/f}LysM^{Cre}* mice were skewed away from immunosuppressive MDSC and displayed features of differentiated neutrophils and properties of neutrophil-mediated immunity.

## SHP-2 deficiency reshapes the intratumoral macrophage infiltrate

Next, we examined the transcriptional properties of TAMs isolated from tumors of *Shp2^{f/f}LysM^{Cre}* and *Shp2^{f/f}* mice at day 15 post-tumor implantation. Differential gene expression analysis showed that among 16,749 expressed genes, a total of 7,307 genes (3,650 upregulated and

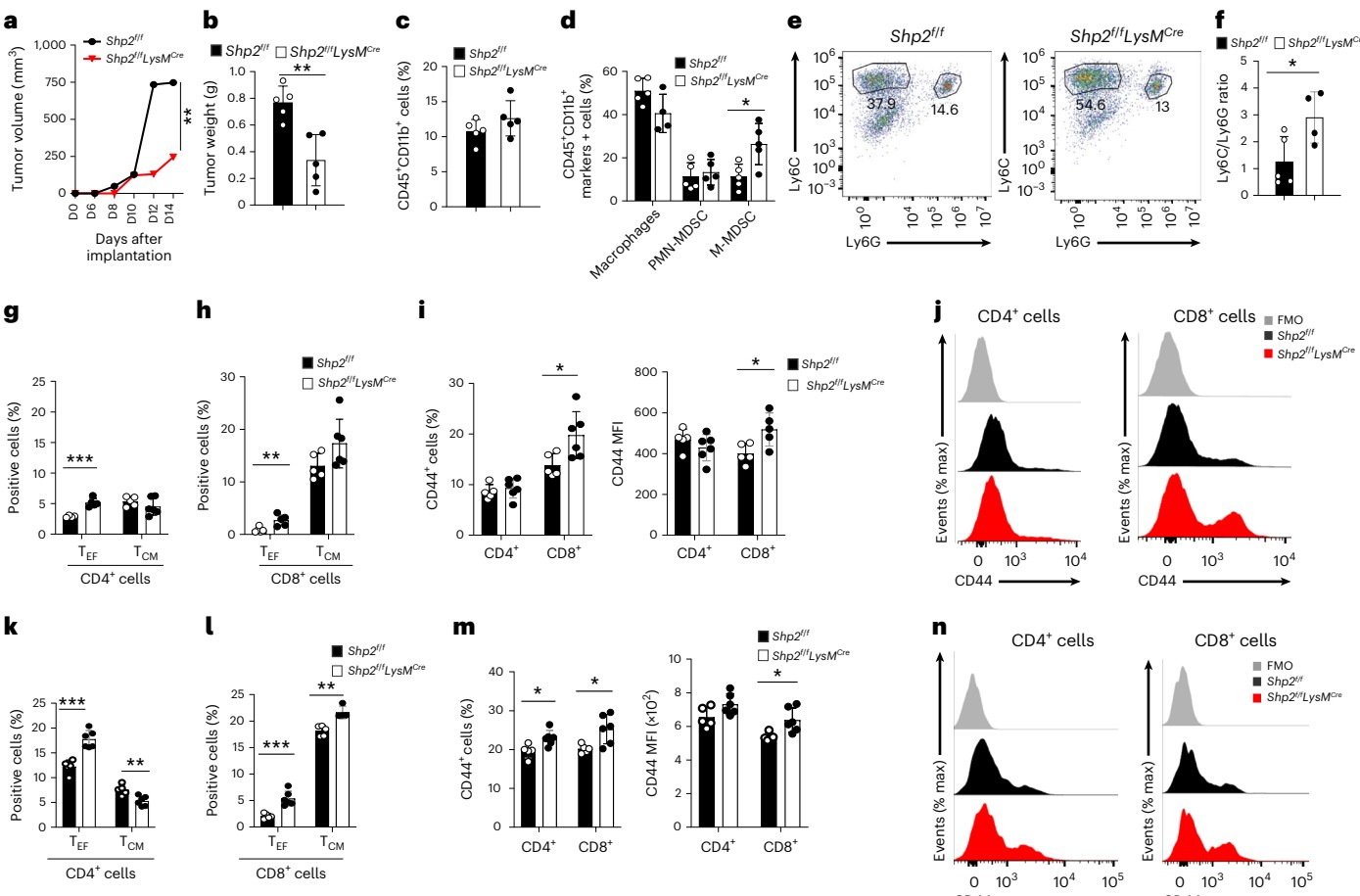

**Fig. 3 | Myeloid-specific SHP-2 deletion alters the MDSC. a,b,** In *Shp2^{f/f}* and *Shp2^{f/f}LysM^{Cre}* mice, tumor volume was monitored for 14 days (**a**) and tumor weight was assessed or on day 14 (**b**) post inoculation with MC17-51 cells. **c–e,** Percentage of CD45⁺CD11b⁺ cells (**c**), macrophages, CD11b⁺Ly6C^{hi}Ly6G⁻ monocytic MDSCs (M-MDSCs) and CD11b⁺Ly6C^{lo}Ly6G⁺ PMN-MDSCs (**d**), and representative pseudocolor plots (**e**) in tumors from mice as in **a. f,** The ratio of M-MDSC/PMN-MDSC in the tumors of *Shp2^{f/f}* and *Shp2^{f/f}LysM^{Cre}* tumor-bearing mice as in **a. g–n,** Frequency of CD4⁺ T_{EF} cells (**g, k**) and CD8⁺ T_{EF} cells (**h, l**) and representative flow cytometry (**j, n**) and quantification (**i, m**) of CD44 expression in CD4⁺ and CD8⁺ T_{EF} cells in dLN (**g–j**), and the spleen (**k–n**) of tumor-bearing mice as in **a.** Mean percentage ± s.d. are shown. Results are representative of three separate experiments with four mice per group and two experiments with seven mice per group. *$P$ = 0.0166–0.0293, **$P$ = 0.0019-0.0085, ***$P$ = 0.0003-0.00085, unpaired two-sided $t$ test.

3,657 downregulated) were differentially expressed between TAMs from *Shp2^{f/f}LysM^{Cre}* and *Shp2^{f/f}* tumor-bearing mice (Fig. 5a,b, Supplementary Table 2, Extended Data Fig. 2b and Supplementary Fig. 5). TAMs from *Shp2^{f/f}LysM^{Cre}* mice had enhanced expression of genes with important roles in monocyte, macrophage or DC differentiation and function, such as the transcription factors *Irf8, Klf4* and *Zeb2* (Fig. 5b,c and Supplementary Table 2), monocyte signature genes (*Cfp, Ly86* and *Csf1R*; Supplementary Table 2) and DC specification genes, such as the transcription factor *Baft3* (Fig. 5b,c and Supplementary Table 2), which together with *Irf8* is required for DC differentiation[21], and the immunostimulatory molecules *CD86* and *CD83* and *Csf1, Siglec1, Clec10* and *Slamf7* (Fig. 5b and Supplementary Table 2), which have been associated with moDC differentiation[22,23]. Although our experimental system was not geared to investigate DC differentiation, these findings indicated that TAMs in *Shp2^{fl/fl}LysM^{Cre}* mice had enhanced gene expression programs identifying monocytes differentiated from MDPs, which can give rise to Ly6C⁺ classical and moDC-producing monocytes and DC[24]. We noted increased expression of *Nos2* (Fig. 5b), which characterizes inflammatory macrophages differentiated from Ly6C^{hi} monocytes[25]; *STING* (encoded by *Tmem173*) (Supplementary Table 2), a pattern recognition receptor that transmits signals activating IFN type I responses[26]; IFN type I-induced proinflammatory genes *Ifit2, Ifit3* and *Cmpk2* (Fig. 5b); and *TLRs* and TLR downstream signaling mediators

such as *Unc93b1* and *Wdfy1* (Fig. 5b,c and Supplementary Table 2) in TAMs of *Shp2^{fl/fl}LysM^{Cre}* tumor-bearing mice, indicating an enhanced proinflammatory program. Conversely, compared to TAMs isolated from *Shp2^{fl/fl}* mice, TAMs of *Shp2^{fl/fl}LysM^{Cre}* had diminished expression of inhibitory genes such as *Havcr2* (encoding for Tim3), *Prdm1, Trem2* and *Wnt* (Fig. 5b,c), all of which have detrimental immunosuppressive roles in myelocyte-mediated antitumor function[19,27,28].

Pathway enrichment analysis and GSEA showed that TAMs from *Shp2^{fl/fl}LysM^{Cre}* mice were enriched for genes of macrophage differentiation and activation, phagocytosis, TLR and NF-kB signaling, cytokine and chemokine activity, IL-1 response, cell killing, antigen-presenting function and autophagy (Fig. 5d,e and Extended Data Fig. 2b), which are associated with antitumor properties of TAMs[29]. The differentially expressed cytokine production pathways included enhanced production of IL-10, IL-17, type I IFN and IFN-γ (Fig. 5f,g), proinflammatory IL-6, IL-1α, IL-1β and IL-18 (Fig. 5h and Supplementary Fig. 2b). Several chemokines—including *Pf4* (*CXCL4*), which acts as a chemoattractant for neutrophils and monocytes and enhances T memory cell responses and *CXL2* and *CXL3*, which collectively promote monocyte, DC, NK and T cell recruitment and macrophage activation, thereby mediating a proinflammatory immune response—were upregulated in *Shp2^{fl/fl}LysM^{Cre}* TAMs compared to *Shp2^{fl/fl}* TAMs (Extended Data Fig. 2b and Supplementary Table 2). TAMs from *Shp2^{f/f}LysM^{Cre}* mice had an

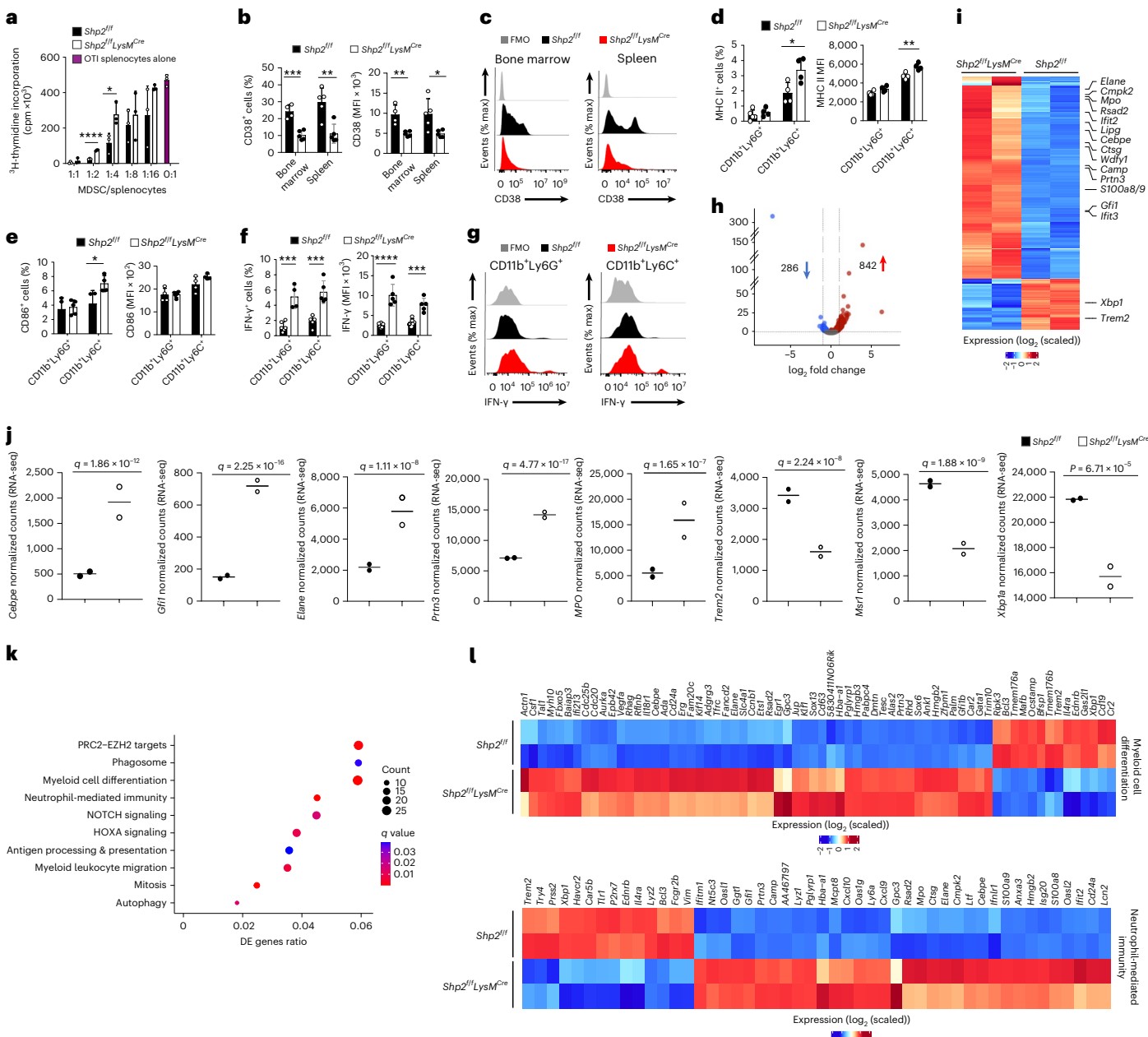

**Fig. 4 | SHP-2 deletion promotes myeloid cell differentiation to mature leukocytes with enhanced neutrophil-mediated immunity. a**, [3]H-thymidine incorporation (cpm) in OTI splenocytes stimulated with OVA$_{257-264}$ cocultured with PMN-MDSC isolated from *Shp2^f/f* and *Shp2^f/f LysM^Cre* mice at day 15 post inoculation of MC17-51 tumor cells. Mean ± s.d. of cpm values is shown. ****$P < 0.0001$ cpm counts obtained by addition of *Shp2^f/f* versus *Shp2^f/f LysM^Cre* PMN-MDSC, unpaired *t* test. Results are representative of three separate experiments using 8–10 mice per group, and three technical replicates per condition. **b,c**, Quantification (**b**) and representative histograms (**c**) of CD38 expression in PMN-MDSC from spleen and BM in mice as in **a**. Mean percentage ± s.d. **d–g**, Quantification of MHC II (**d**) and CD86 (**e**) expression in tumor-infiltrating CD11b⁺Ly6C^loLy6G⁺ and CD11b⁺Ly6C^hiLy6G⁻ cells and quantification (**f**) and representative histograms (**g**) of IFN-γ expression in tumor-infiltrating CD11b⁺Ly6C^loLy6G⁺ and CD11b⁺Ly6C^hiLy6G⁻ cells in mice as in **a**. Results are from one of three independent experiments with *n* = 5 mice per group. **h,i**, Volcano plot of differentially expressed (DE) genes (**h**) and heat map (**i**) of top 600 DE genes in PMN-MDSC cells isolated from spleens of *Shp2^f/f* and *Shp2^f/f LysM^Cre* tumor-bearing mice and analyzed by bulk RNA-seq (*q* < 0.05 for all DEGs, log₂(FC) > 0 and log₂(FC) < 0 for upregulated and downregulated genes, respectively). **j**, Expression of the indicated genes in PMN-MDSC from *Shp2^f/f* and *Shp2^fl/fl LysM^Cre* tumor-bearing mice (data from RNA-seq dataset). **k**, Bubble plot of substantially enriched pathways (*q* < 0.1) in cells from *Shp2^fl/fl LysM^Cre* mice sorted by GeneRatio. **l**, Heat maps of differentially expressed genes involved in myeloid cell differentiation and neutrophil-mediated immunity (*q* < 0.05). *$P = 0.026$, ***$P = 0.0034–0.004$, ****$P < 0.0001$, unpaired *t* test.

enhanced signature of multiple metabolic pathways, including lipid, carbohydrate and amino acid transport, cholesterol metabolism and energy metabolism (Fig. 5i). Enhanced metabolic activity, characterized by glucose and glutamine metabolism, increased levels of amino acids and anabolic lipid metabolism were also observed in phagocytes generated from the bone marrow of *Shp2^f/f LysM^Cre* and *Shp2^f/f* mice by culture with GM-CSF (Extended Data Fig. 3). Consistent with these findings, GSEA using Gene Ontology Biological Processes Pathways gene sets showed that TAMs from *Shp2^f/f LysM^Cre* mice were characterized by highly expressed signatures of leukocyte activation, chemotaxis, migration, cytokine production and inflammatory response (Extended Data Fig. 4a).

To examine whether the properties of myeloid cells from *Shp2^f/fLysM^Cre* mice had a direct role in the improved antitumor responses, we examined antitumor responses after T cell depletion in *Shp2^f/f* and *Shp2^f/fLysM^Cre* mice. T cell depletion increased tumor growth in *Shp2^f/f* mice but only marginally increased tumor growth in *Shp2^f/fLysM^Cre* mice, indicating that non-T cell types had a more prominent role in mediating antitumor responses in *Shp2^f/fLysM^Cre* mice (Extended Data Fig. 4b,c). These results indicated that myeloid cells from *Shp2^f/fLysM^Cre* tumor-bearing mice developed a program of enhanced proinflammatory differentiation and features consistent with enhanced antigen processing and presentation leading to T cell activation but also had an ability to mediate T cell-independent antitumor function.

## SHP-2 ablation induces lasting antitumor properties in monocytes

Next, we investigated whether the neutrophils and monocytes in tumor-bearing *Shp2^f/fLysM^Cre* mice mediated lasting antitumor protection indicative of trained immunity. At day 9 post MC17-51 tumor injection, a time at which tumors had similar size in *Shp2^f/f* and *Shp2^f/fLysM^Cre* mice, we collected CD45+CD11b+Ly6C^hiLy6G− monocytes and CD45+CD11b+Ly6C^loLy6G+ neutrophils from the bone marrow of tumor-bearing *Shp2^f/fLysM^Cre* mice, mixed them with an equal number of MC17-51 tumor cells and injected them subcutaneously into naïve wild-type (WT) mice[30]. There was no difference in tumor growth in WT mice that received neutrophils from *Shp2^f/fLysM^Cre* tumor-bearing mice compared to mice that were injected with MC17-51 tumors alone, whereas WT mice that received monocytes from *Shp2^f/fLysM^Cre* tumor-bearing mice had substantially reduced tumor growth compared to those that were injected with MC17-51 tumors alone (Fig. 6a,b). These findings indicated that bone marrow CD45+CD11b+Ly6C^hiLy6G− monocytes in *Shp2^f/fLysM^Cre* mice had an antitumor function that could be transferred to new hosts.

Development of trained immunity in other models has been associated with an expansion of distinct types of hematopoietic progenitors or mature myeloid cell subsets[31]. Quantification of distinct subsets of myeloid progenitors and mature myeloid cells in the bone marrow of tumor-bearing *Shp2^f/f* and *Shp2^f/fLysM^Cre* mice at day 9 post implantation of MC17-51 tumors, before isolation of monocytes and neutrophils for adoptive transfer, showed similar number of Lin− myeloid progenitors including Flt3+CD115^lo CMP, FLT3−CMP, MDP, GMP, GP, MP+cMoP or mature differentiated Lin+ myeloid cells, including total CD45+CD11b+ myeloid cells, CD11b+Ly6C^hiLy6G− monocytes and CD11b+Ly6C^loLy6G+ granulocytes at this timepoint (Fig. 6c,d and Extended Data Fig. 5). These observations indicated that bone marrow monocytes in *Shp2^f/fLysM^Cre* tumor-bearing mice were imprinted with antitumor properties, while the number of bone marrow myeloid progenitors were not altered.

## SHP-2 and PD-1-SHP-2 signaling impede phosphorylation of HOXA10 and IRF8

Next, we examined whether phosphorylation of the SHP-2 targets HOXA10 and IRF8 was altered in SHP-2 deficient bone marrow myeloid cells. Flow cytometric analysis of bone marrow Lin− myeloid progenitor cells cultured with GM-CSF+IL-3 for 3 days indicated that the percentage of Lin− myeloid progenitors had decreased while that of Lin+ cells had increased (Extended Data Fig. 6a), and ≥ 95% of the Lin+ cells were CD45+CD11b+ myelocytes (Extended Data Fig. 6b), indicating differentiation of myeloid progenitors. In cell lysates from *Shp2^f/fLysM^Cre* and *Shp2^f/f* bone marrow cultured for 48 h with GM-CSF+IL-3, HOXA10 or IRF8 immunoprecipitation followed by immunoblot with a phosphotyrosine antibody showed enhanced HOXA10 or IRF8 phosphorylation in myelocytes from *Shp2^f/fLysM^Cre* myelocytes compared to *Shp2^f/f* (Fig. 7a,b). When cultured with GM-CSF with or without IL-3, myeloid progenitors and their progeny express PD-1 and PD-L1 (Extended Data Fig. 6c,d)[7]. PD-1 immunoprecipitation followed by SHP-2 immunoblot

detected a robust PD-1-SHP-2 interaction in GM-CSF+IL-3-cultured myelocytes from *Shp2^f/f* but not *Shp2^f/fLysM^Cre* mice (Fig. 7a,b).

Erk and mTOR are targets of PD-1 in T cells[32], and the GM-CSF-mediated activation of Erk and mTOR was reported to be enhanced in PD-1-deficient CMPs and GMPs during culture[7]. During culture of primary bone marrow, PD-1 and PD-L1 were expressed at low levels in both Lin− and Lin+ cells before treatment with GM-CSF+IL-3 and were upregulated after treatment (Fig. 7e and Extended Data Fig. 6c,d). In T cells, SHP-2 is recruited to the cytoplasmic tail of PD-1 after tyrosine phosphorylation by TCR-mediated activation of Src kinases Fyn and Lck[33,34]. In the myeloid lineage, the βc subunit of the GM-CSF receptor (GM-CSFR) represents a major signaling subunit and is tyrosine phosphorylated in response to cytokine stimulation[35]. The Src kinase Lyn can directly associate with GM-CSFR βc subunit. Immunoprecipitation with PD-1 antibody followed by immunoblot with phospho-specific PD-1 antibody, which recognizes pY248, a site within the conserved ITSM motif of PD-1 cytoplasmic tail known to be phosphorylated by Src family kinases leading to PD-1 interaction with SHP-2 in T cells[33,34], showed that GM-CSF induced pY248 and PD-1 interaction with SHP-2 (Fig. 7d,e). Sequential immunoblot with GM-CSFRβc-specific and Lyn-specific antibodies showed that both proteins were detected in PD-1 immunoprecipitates (Fig. 7d,e). Thus, GM-CSF-mediated signaling in myeloid cells induced the recruitment of PD-1 and SHP-2 phosphatase to GM-CSFRβc, a major signaling receptor involved in myelocyte activation, proliferation and differentiation.

Phosphorylation of HOXA10 and IRF8 was increased in whole bone marrow cells from *Pdcd1^f/fLysM^Cre* mice, in which the *Pdcd1* gene (encoding PD-1) is deleted selectively in the myeloid compartment, cultured with GM-CSF+IL-3 for 48 h (Fig. 7f,g) or in WT bone marrow cells cultured with GM-CSF+IL-3 for 48 h in the presence of a PD-L1 blocking antibody, but not in the presence of an isotype-matched control antibody (Extended Data Fig. 7). These results indicated that PD-1-SHP-2 signaling suppressed HOXA10 and IRF8 phosphorylation, which have important roles in myeloid differentiation and lineage fate commitment[9,13].

## PD-1 ablation alters the properties of monocytes and TAMs

SHP-2 has multiple interactors but is the only direct interacting partner for PD-1 identified to date[36]. Next, we examined whether the myeloid-specific PD-1 deletion induced the generation of monocytes with antitumor properties and molecular signatures similar to SHP-2 deficiency. *Pdcd1^f/fLysM^Cre* mice had smaller tumors during longitudinal monitoring after implantation with MC17-51 tumor cells compared to control *Pdcd1^f/f* mice (Extended Data Fig. 8a,b), consistent with previous observations in different tumor models[7]. The percentage of IRF8+ M-MDSC (Extended Data Fig. 8c) and TAMs (Extended Data Fig. 8d) and the expression of CD80 and CD86 in TAMs (Extended Data Fig. 8e) was increased in *Pdcd1^f/fLysM^Cre* tumor-bearing mice compared to *Pdcd1^f/f* tumor-bearing mice.

To determine whether PD-1 deficient monocytes had the ability to develop trained immunity and lasting antitumor protection, CD45+CD11b+Ly6C^hiLy6G− monocytes isolated from the bone marrow of *Pdcd1^fl/flLysM^Cre* or *Pdcd1^f/f* mice on day 9 post injection with MC17-51 tumor cells, when tumors had comparable size between the two groups, were mixed with an equal number of MC17-51 tumor cells and injected subcutaneously into naïve WT mice. There was no difference in tumor growth between WT mice implanted with MC17-51 tumor cells alone and recipients of monocytes from tumor-bearing *Pdcd1^f/f* mice, whereas recipients of monocytes from tumor-bearing *Pdcd1^f/fLysM^Cre* mice had substantially reduced tumor growth (Fig. 8a,b), indicating that PD-1-deficient monocytes could mediate tumor control. Similar numbers of bone marrow Lin− myeloid progenitors, including Flt3+CD115^lo CMP, FLT3−CMP, MDP, GMP, GP and MP+cMoP, or mature differentiated Lin+ myeloid cells, including total CD45+CD11b+ myeloid cells, CD11b+Ly6C^hiLy6G− monocytes and CD11b+Ly6C^loLy6G+ granulocytes,

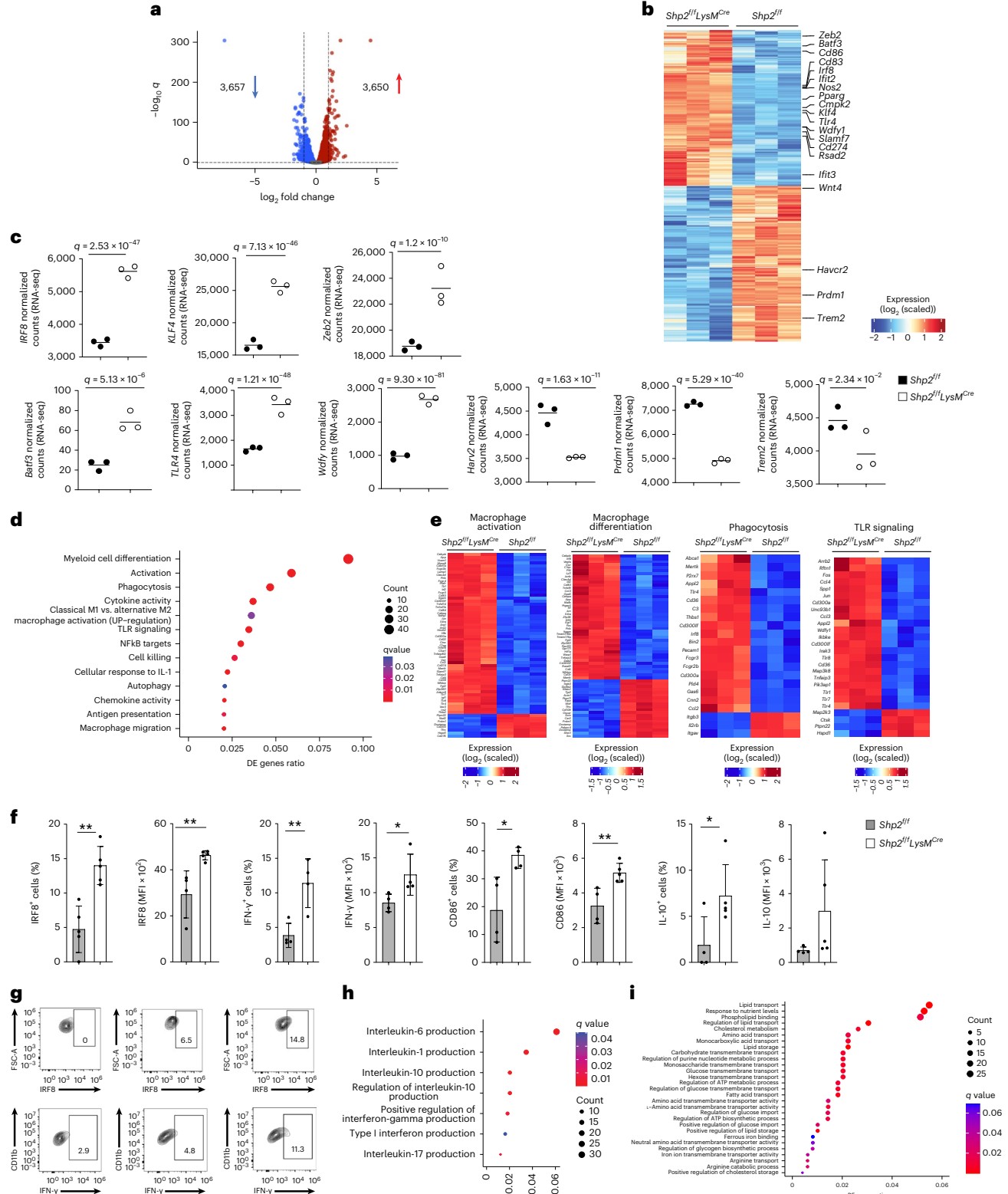

**Fig. 5 | SHP-2 deletion increases monocyte and DC specification gene transcripts and imprints an effector differentiation program in TAMs.**
**a,b,** Volcano plot of DE genes (**a**) and heat map (**b**) of the top 6,500 DE genes in TAMs isolated from MC17-51 tumor-bearing *Shp2^f/f* and *Shp2^f/f*LysM^Cre* mice and analyzed by RNA-seq. log₂(FC) > 0 and log₂(FC) < 0 for upregulated and downregulated genes, *q* < 0.05 for all DEGs, respectively). **c,** Expression of the indicated genes by TAMs from *Shp2^f/f* and *Shp2^f/f*LysM^Cre* tumor-bearing mice (data from RNA-seq dataset). **d,** Bubble plot of substantially enriched pathways (*q* < 0.1) in TAMs of *Shp2^f/f*LysM^Cre* mice sorted by GeneRatio. **e,** Heat maps of

differentially expressed genes related to macrophage differentiation, macrophage activation, phagocytosis and TLR signaling in TAMs from *Shp2^f/f* and *Shp2^f/f*LysM^Cre* tumor-bearing mice (*q* < 0.05). **f,g,** Quantification of IRF8, IFN-γ, CD86 and IL-10 expression (**f**) and representative flow cytometry of IRF8 and IFN-γ expression (**g**) in TAMs of tumor-bearing *Shp2^f/f* and *Shp2^f/f*LysM^Cre* mice. Results are representative of four independent experiments with four to six mice per group. **h,i,** Bubble plot of cytokine pathways (**h**) and metabolism pathways (**i**) substantially enriched in TAMs (*q* < 0.1) from *Shp2^f/f*LysM^Cre* mice sorted by GeneRatio.

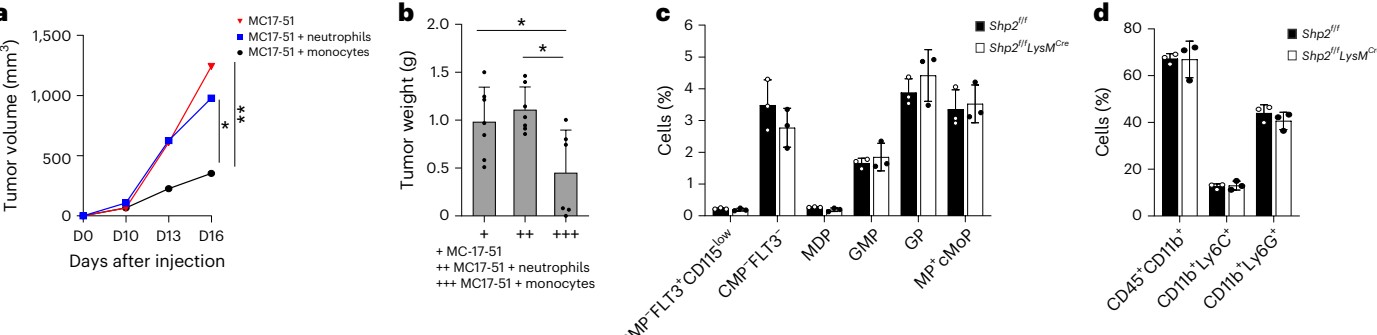

**Fig. 6 | SHP-2 ablation induces lasting antitumor properties in monocytes.** **a**, Tumor size in naïve WT mice subcutaneously injected with MC17-51 tumor cells with or without injection with CD45⁺CD11b⁺Ly6CʰⁱLy6G⁻ monocytes or CD45⁺CD11b⁺Ly6C⁻Ly6G⁺ neutrophils isolated from the bone marrow of *Shp2ᶠ/fLysMᶜʳᵉ* mice on day 9 post inoculation of MC17-51 tumors. **b**, Tumor weight on day 15 in mice as in **a**. Results show mean + s.d. Results are from one of two experiments with *n* = 10 mice per group. *P = 0.0104–0.0387, **P = 0.0033, ANOVA. **c**,**d**, The indicated subsets of Lin⁻ myeloid progenitors (**c**) and mature CD45⁺CD11b⁺ myeloid cells and the subsets of CD45⁺CD11b⁺Ly6CʰⁱLy6G⁻ monocytes (CD11b⁺Ly6C⁺) and CD45⁺CD11b⁺Ly6C⁻Ly6G⁺ neutrophils (CD11b⁺Ly6C⁺) were assessed in the bone marrow of *Shp2ᶠ/f* and *Shp2ᶠ/fLysMᶜʳᵉ* tumor-bearing mice on day 9 before collection of monocytes or neutrophils from *Shp2ᶠ/fLysMᶜʳᵉ* tumor-bearing mice for transfer into the new hosts. Results are from one of three separate experiments with *n* = 5 mice per group.

were detected in *Pdcd1ᶠ/fLysMᶜʳᵉ* and *Pdcd1ᶠ/f* tumor-bearing mice at day 9 post-tumor inoculation (Extended Data Fig. 8f,g).

Next, we performed RNA-seq in TAMs isolated from *Pdcd1ᶠ/fLysMᶜʳᵉ* and *Pdcd1ᶠ/f* tumor-bearing mice. Differential gene expression analysis showed that among 16,552 expressed genes, a total of 1,766 genes (846 upregulated and 920 downregulated) were differentially expressed between TAMs from *Pdcd1ᶠ/fLysMᶜʳᵉ* and *Pdcd1ᶠ/f* mice (Fig. 8c,d and Supplementary Table 3). *Pdcd1ᶠ/fLysMᶜʳᵉ* TAMs had enhanced expression of genes consistent with myeloid cell differentiation, such as *Myadm*; inflammatory activation, such as *Klf7*, the dectin-2 family of C-type lectin receptors *Clec4e, Clec4d* and *Clec4n*; and the paracaspase *Malt1* (Fig. 8d,e and Supplementary Table 3), which together with Clec4n are indispensable determinants of responses to PD-1 blocking immunotherapy[37]. There was also increased expression of *Itgax* (CD11c), *Nr4a2, Nr4a3, Egr2, Egr3, Bhlhe40* and *CD24* transcripts, which are related to DC maturation[21] (Fig. 8d,e and Supplementary Table 3), indicating that *Pdcd1ᶠ/fLysMᶜʳᵉ* TAMs were derived from monocytes differentiated from MDPs that could give rise to Ly6C⁺ classical and moDC-producing monocytes, and DC[24].

Consistent with an activated macrophage profile, there was an increase of *CD80, CD86* (Fig. 8d and Supplementary Table 3), inflammatory mediators, including the TLR downstream mediators *Wdfy1* and *CD180* (Fig. 8d,e, Supplementary Table 3 and Extended Data Fig. 9a), and multiple proinflammatory cytokines, including *IL-6, IL-1a, IL-1b* and *IL-23a* (Fig. 8d and Supplementary Table 3). Conversely, there was a diminished expression of genes that convey immunosuppressive functions of TAMs such as *Trem2, Mertk, CD163* and *Mrc1* (CD206) (Fig. 8d,e and Supplementary Table 3). Compared to *Pdcd1ᶠ/f* TAMs, *Pdcd1ᶠ/fLysMᶜʳᵉ* TAMs had increased expression of *Jarid2* (Fig. 8d), a histone methyltransferase acting as an accessory subunit for the core PRC, recruiting PRC2 complex to target genes and epigenetically regulating gene expression[38]. *Pdcd1ᶠ/fLysMᶜʳᵉ* TAMs had increased expression of *Hif1a* (Fig. 8d and Supplementary Table 3), which promotes glycolysis under hypoxia but also serves as an indispensable metabolic mediator of trained immunity[39] and pyruvate dehydrogenase phosphatase 1 (*Pdp1*) (Fig. 8d,e and Supplementary Table 3), which regulates the activation of pyruvate dehydrogenase complex converting pyruvate to acetyl-CoA for entry to the TCA cycle and synthesis of itaconate, serving a critical metabolic step required by proinflammatory macrophages to sustain cytokine production[40]. In parallel, there was decreased expression of *Cpt1a*, which regulates mitochondrial entry of long-chain fatty acids promoting fatty acid oxidation, and a concomitant increase of *Hmgcs1* (Fig. 8d,e and Supplementary Table 3), the enzyme catalyzing

condensation of acetyl-CoA with acetoacetyl-CoA to form HMG⁻CoA that is converted into mevalonate, the precursor of cholesterol synthesis and a mediator of trained immunity[41].

GSEA of the top 500 differentially expressed genes showed that *Pdcd1ᶠ/fLysMᶜʳᵉ* TAMs were highly enriched for genes involved in leukocyte activation and differentiation, chemotaxis, protein secretion and inflammatory response, phagocytosis, cytokine and chemokine activity, DC differentiation and maturation, and response to type I IFNs (Fig. 8f and Extended Data Fig. 9a), all functions associated with antitumor properties of TAMs[42,43]. In addition, there was high enrichment for genes of signaling pathways, including the MAPK cascade, kinase activity, Erk1/2 cascade, Ras and second messenger medicated signaling, NF-kB, calcium ion transport, PI3K, PKB, phospholipase activity and tyrosine kinase activity (Fig. 8f), signaling pathways targeted by PD-1 in T cells[32]. *Pdcd1ᶠ/fLysMᶜʳᵉ* TAMs were also highly enriched for genes involved in oxidative stress response, phospholipid binding and lipid transport, glucose homeostasis, amino acid metabolism and fatty acid biosynthetic processes (Fig. 8f). Thus, myeloid-specific PD-1 ablation resulted in the generation of TAMs with enhanced signaling and anabolic metabolism.

Comparison of gene expression and GSEA in transcriptomics indicated multiple genes displaying similar changes in *Shp2ᶠ/fLysMᶜʳᵉ* and *Pdcd1ᶠ/fLysMᶜʳᵉ* TAMs (Fig. 8d and Supplementary Table 4). Among the commonly upregulated genes were macrophage activation markers such as *Myadm, Itgax*, the C-type lectin receptors *Clec4e, Clec4d* and *Clec4n*, type I IFN-induced genes such as *Ifi205, Rsad2* and *Ifit1*, the small GTPase *Rap1a* and the Rap1-interacting partner *RIAM (Apbb1ip)*, which has an indispensable role in phagocytosis[44]; TLR signaling mediators such as *Wdfy1* and *CD180*, macrophage activation genes such as *KLF7* and *CD86*, and metabolic regulators including *Hif1a, Pdp1* and *Hmgcs1* (Fig. 8d and Supplementary Table 4).

Among pathways enriched in the differentially upregulated genes in *Shp2ᶠ/fLysMᶜʳᵉ* TAMs, 22% overlapped with pathways enriched in *Pdcd1ᶠ/fLysMᶜʳᵉ* TAMs (Fig. 8g), consistent with multiple signaling interactions of SHP-2 besides PD-1. Conversely, 59% of the pathways enriched in *Pdcd1ᶠ/fLysMᶜʳᵉ* TAMs overlapped with pathways enriched in *Shp2ᶠ/fLysMᶜʳᵉ* TAMs (Fig. 8g), indicating that the majority of PD-1-mediated functions in TAMs were mediated through SHP-2. Common functional pathways enriched in both datasets included genes involved in myeloid cell activation, chemotaxis, proliferation and differentiation, synapse organization and cell adhesion, leukocyte-mediated immunity, TLR signaling and production of inflammatory and effector cytokines (Fig. 8h). Common signaling pathways were also enriched

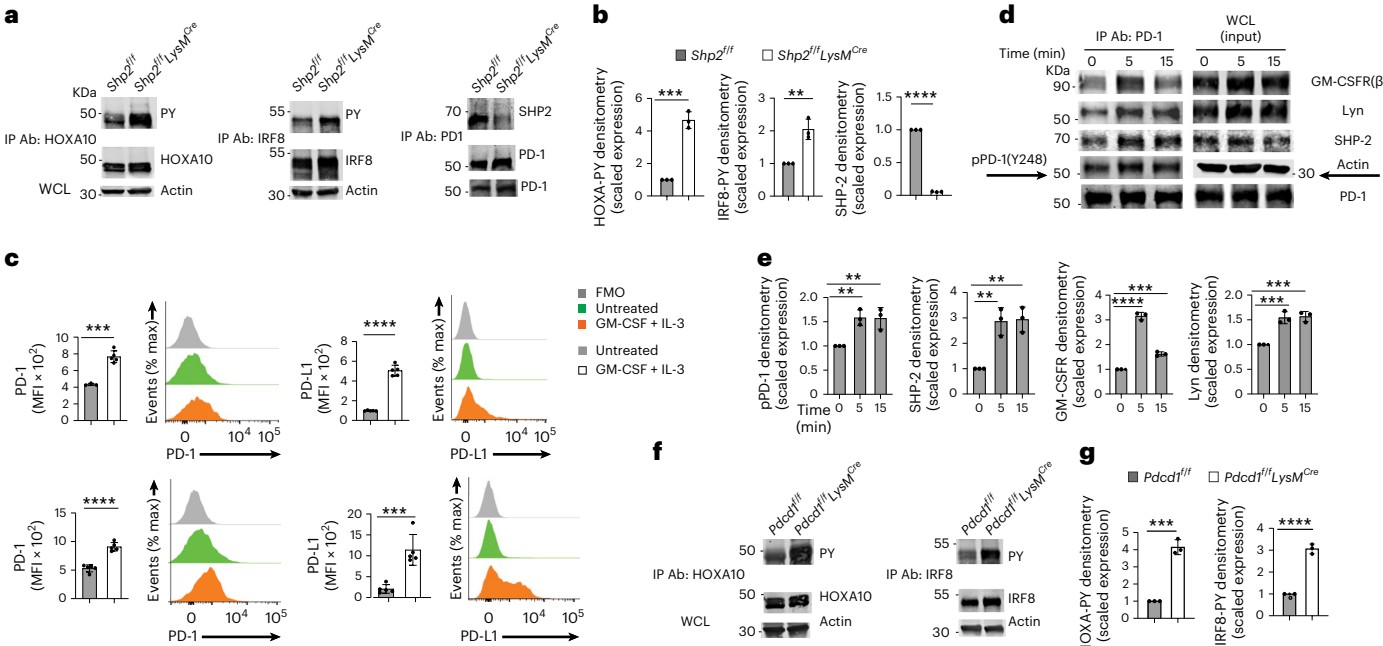

**Fig. 7 | Ablation of SHP-2 or PD-1 enhances GMC-SF-mediated phosphorylation of HOXA1- and IRF8. a**, Immunoprecipitation with agarose-conjugated antibodies specific for HOXA10, IRF8 or PD-1 followed by SDS-PAGE and immunoblot with antibodies specific for pY or HOXA10, IRF8 or PD-1, respectively, in cell lysates from *Shp2*[f/f] and *Shp2*[f/f]*LysM*[Cre] bone marrow cells cultured for 48 h with GM-CSF (10 ng ml⁻¹) and IL-3 (5 ng ml⁻¹). Expression of actin in whole-cell lysates was examined as input. **b**, Abundance of phosphorylated HOXA10, IRF8 or PD-1 normalized to immunoprecipitated HOXA10, IRF8 or PD-1 and expressed as fold change over the value obtained in *Shp2*[f/f] cells, defined as one. Results are representative of three experiments. **c**, Expression of PD-1 and PD-L1 in Lin⁻ (top) and Lin⁺ (bottom) cells following 48 h of the bone marrow from C57BL/6 WT mice as in **a**. MFI ± s.d. and representative histograms are shown. Results are from one of five experiments with four to six mice per group. **d**, Immunoprecipitation with agarose-conjugated PD-1 antibody followed by SDS-PAGE and immunoblot with the indicated antibodies in cell lysates from

C57BL/6 WT bone marrow cells cultured as in **a**, rested for 3 h and then either left untreated or stimulated with GM-CSF (40 ng ml⁻¹) for the indicated time points. **e**, The abundance of PD-1 phosphorylated at Y248 (pPD-1), SHP-2, GM-CSFR(βc) and Lyn coprecipitated with PD-1 from the cell lysates was normalized to immunoprecipitated PD-1 and was expressed as fold change over the value obtained in nonstimulated cells at the zero timepoint (defined as one). Expression of indicated proteins in whole-cell lysates was also examined. Results from one of two experiments are shown. **f**, Immunoprecipitation with agarose-conjugated HOXA10-specific antibody or agarose-conjugated IRF8-specific antibodies followed by SDS-PAGE and immunoblot with antibodies specific for pY followed by immunoblot with HOXA10 or Irf8 in bone marrow cells from *Pdcd1*[f/f] and *Pdcd1*[f/f]*LysM*[Cre] mice cultured as in **a**. Expression of actin in whole-cell lysates was examined as input. **g**, Quantification of phosphorylated HOXA10 and IRF8 was assessed as in **b**. Results are from one of two experiments are shown (**f,g**). **P = 0.0025–0.0084, ***P = 0.0004–0.0084, ****P < 0.0001, t-test.

in the gene sets of *Pdcd1*[f/f]*LysM*[Cre] and *Shp2*[f/f]*LysM*[Cre] TAMs, including Ras pathway activation, kinase activation, MAPK and Erk1/2 cascade, second messenger-mediated signaling, calcium ion homeostasis, calcium-mediated signaling and NF-kB activation (Extended Data Fig. 9b). Thus, TAMs in *Pdcd1*[fl/fl]*LysM*[Cre] and *Shp2*[fl/fl]*LysM*[Cre] mice were governed by molecular mediators that converged in multiple common signaling pathways and biological processes.

IRF8 programs the differentiation of monocytes and DC through epigenetic regulation of distinct sets of enhancers in cooperation with other transcription factors[45]. We examined the expression of genes recently identified to be induced by IRF8 in mature phagocytic cells of the monocyte/DC lineage[45]. In TAMs from *Shp2*[fl/fl]*LysM*[Cre] mice, 891 (24%) of the 3,650 upregulated genes were IRF8 targets (Fig. 8i and Supplementary Table 5) and in TAMs from *Pdcd1*[fl/fl]*LysM*[Cre] mice, 221 (26%) of the 846 upregulated genes were IRF8 targets (Fig. 8j and Supplementary Table 5). Common and distinct IRF8-regulated genes were upregulated in *Shp2*[f/f]*LysM*[Cre] and *Pdcd1*[f/f]*LysM*[Cre] TAMs (Fig. 8i,j) consistent with the ability of IRF8 to cooperate with common and distinct transcription factors in different cells, based on differential concentrations of IRF8 and the differential presence of cooperating transcription factors[45]. Among the common IRF8 targets upregulated in *Shp2*[f/f]*LysM*[Cre] and *Pdcd1*[f/f]*LysM*[Cre] TAMs were the scavenger receptor *CD36*, the costimulatory molecule *CD86*, the volume-regulated anion channel *Lrrc8*c, which regulates STING activation and production of type I interferons, the enzyme *Hmgcs1*, the transcription factor *Egr2*,

the antioxidant mediator *Tmx4*, macrophage activation genes such as *Vcan* and *Clec4n* and the cytokine IL-10 (Fig. 8i,j and Supplementary Table 5). Among IRF8-target genes upregulated selectively in *Shp2*[f/f]*LysM*[Cre] TAMs were the transcription factors *Nr4a1*, *Maff* and *Zeb2* (Fig. 8i), whereas among IRF8 targets upregulated specifically in *Pdcd1*[f/f]*LysM*[Cre] TAMs were the costimulatory molecule *CD80*, the member of the tetraspanin family *CD9*, the epigenetic regulator *Jarid2*, and the transcription factor *Cebpe* (Fig. 8j), suggesting enhanced IRF8 function in SHP-2 and PD-1 deficient TAMs. These results showed that TAMs in *Pdcd1*[fl/fl]*LysM*[Cre] mice had a transcriptomic profile associated with inflammatory differentiation and activation, accompanied by enhanced signaling and metabolic reprogramming and considerable enrichment of IRF8-regulated genes, and that molecular signatures of TAMs in *Pdcd1*[fl/fl]*LysM*[Cre] and *Shp2*[fl/fl]*LysM*[Cre] mice converged in multiple common signaling pathways and biological processes.

### IL-10 is involved in the enhanced antitumor immunity

Although considered an immunosuppressive mediator, IL-10 can have a proimmunogenic role with considerable implications in antitumor immunity[46]. Because IL-10, an IRF8-regulated gene[45], was upregulated in TAMs of *Shp2*[fl/fl]*LysM*[Cre] and *Pdcd1*[fl/fl]*LysM*[Cre] mice, we examined its role in the altered immunological properties of myeloid cells and the enhanced antitumor responses in *Shp2*[fl/fl]*LysM*[Cre] and *Pdcd1*[fl/fl]*LysM*[Cre] mice. In WT, *Shp2*[f/f]*LysM*[Cre] and *Pdcd1*[f/f]*LysM*[Cre] mice treated with IL-10 neutralizing antibody or isotype control on day 9, 11 and 13 post injection with

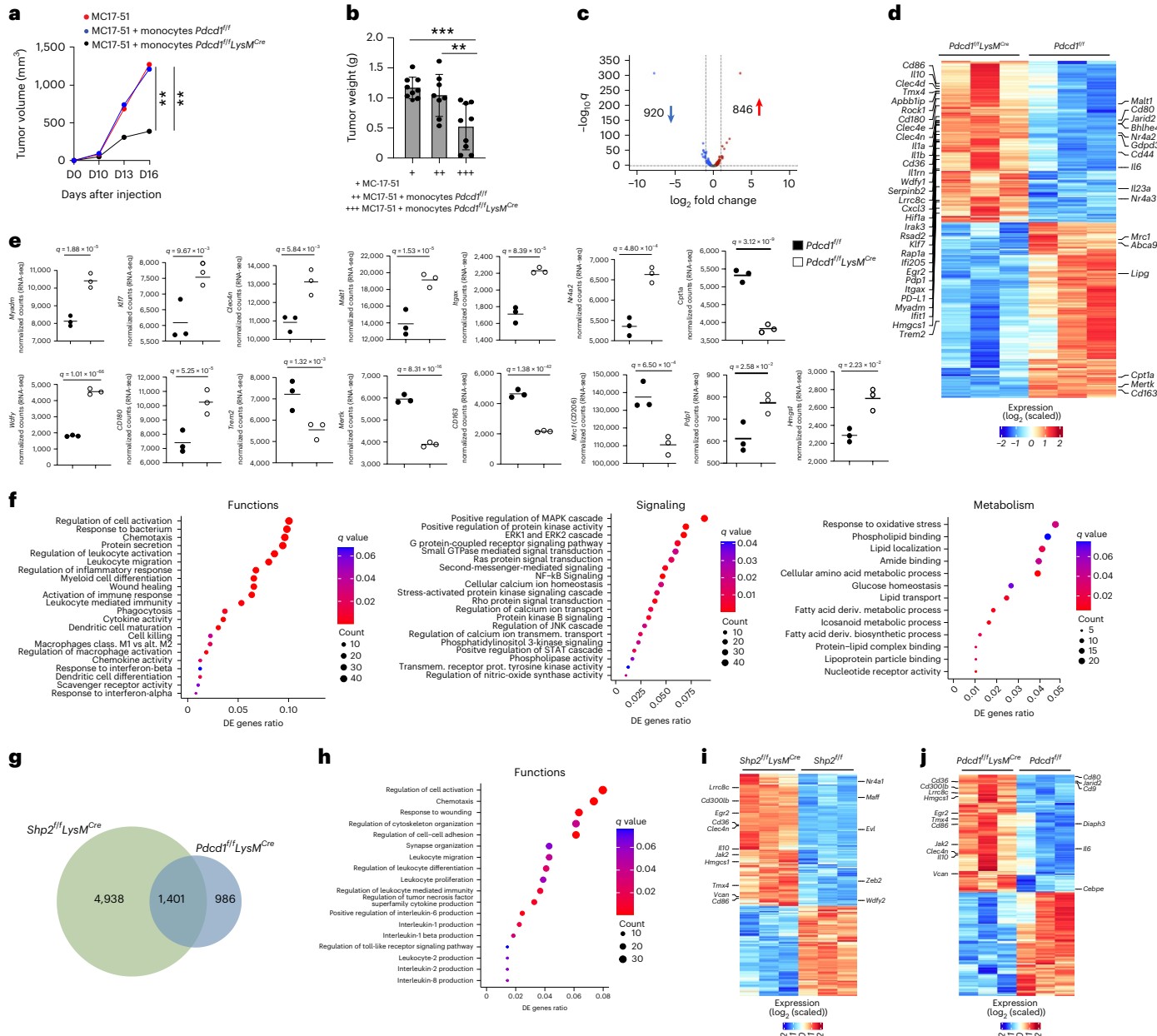

**Fig. 8 | PD-1 deletion altered signaling and metabolism, and imprinted an effector function program in TAMs. a,b,** Tumor size in naïve WT mice subcutaneously injected with MC17-51 tumor cells with or without CD45⁺CD11b⁺Ly6CʰⁱLy6G⁻ monocytes isolated from the bone marrow of *Pdcd1ᶠˡ/ᶠˡLysMᶜʳᵉ* and *Pdcd1ᶠˡ/ᶠˡ* mice on day 9 post inoculation of MC17-51 tumors. **b,** Tumor weight on day 15 in mice as in **a**. Results show mean + s.d. Results are from one of two experiments with $n = 10$ mice per group. \*\*$P = 0.0055–0.0079$, \*\*\*$P = 0.0006$, ANOVA. **c,d,** Volcano plot of DE genes (**c**) and heat map (**d**) of 1,766 genes differentially expressed in TAMs isolated from MC17-51 tumor-bearing *Pdcd1ᶠ/ᶠ* and *Pdcd1ᶠˡ/ᶠˡLysMᶜʳᵉ* mice and analyzed by RNA-seq. $\log_2(FC) > 0$ and $\log_2(FC) < 0$ for upregulated and downregulated genes, $q < 0.05$ for all DEGs, respectively).

**e,** Expression of the indicated genes in TAMs from *Pdcd1ᶠ/ᶠ* and *Pdcd1ᶠˡ/ᶠˡLysMᶜʳᵉ* tumor-bearing mice (data from RNA-seq dataset). **f**. Bubble plot of substantially enriched functional, signaling and metabolic pathways ($q < 0.1$) among the top 500 DE genes in TAMs of *Pdcd1ᶠˡ/ᶠˡLysMᶜʳᵉ* mice compared to *Pdcd1ᶠ/ᶠ* mice sorted by GeneRatio. **g,** Venn Diagrams depicting the overlap of substantially enriched pathways between upregulated DE genes in *Shp2ᶠˡ/ᶠˡLysMᶜʳᵉ* and *Pdcd1ᶠˡ/ᶠˡLysMᶜʳᵉ* TAMs. **h,** Bubble plot of common pathways enriched in among the top 500 DE genes in *Shp2ᶠˡ/ᶠˡLysMᶜʳᵉ* and *Pdcd1ᶠˡ/ᶠˡLysMᶜʳᵉ* TAMs. **i,j.** Heat maps of the top 6,500 DE genes in TAMs of tumor-bearing *Shp2ᶠˡ/ᶠˡLysMᶜʳᵉ* mice (**i**) and heat maps of the top 1,766 DE genes in TAMs of *Pdcd1ᶠˡ/ᶠˡLysMᶜʳᵉ* mice (**j**). Representative common (left) and distinct (right) IRF8-upregulated genes are annotated.

MC17-51 tumor cells, there was no difference in tumor growth between IL-10 Ab-treated and isotype-treated WT mice, while IL-10 Ab-treated *Shp2ᶠˡ/ᶠˡLysMᶜʳᵉ* and *Pdcd1ᶠˡ/ᶠˡLysMᶜʳᵉ* mice had substantially enhanced tumor growth compared to their counterparts treated with isotype control Ig (Extended Data Fig. 10a,b). IL-10 Ab treatment did not alter the diminished suppressor function of MDSCs from *Shp2ᶠˡ/ᶠˡLysMᶜʳᵉ* and *Pdcd1ᶠˡ/ᶠˡLysMᶜʳᵉ* mice (Extended Data Fig. 10c,d). The fractions of

myeloid subsets (Extended Data Fig. 10e) and expression of MHC II, CD80 or CD86 were similar in TAMs from IL-10 Ab- or isotype-treated *Shp2ᶠ/ᶠLysMᶜʳᵉ* or *Pdcd1ᶠˡ/ᶠˡLysMᶜʳᵉ* mice (Extended Data Fig. 10f), indicating that IL-10 was not responsible for the diminished suppressor function of SHP-2- or PD-1-deficient myeloid cells in this experimental system. However, IL-10 had an active role in the enhanced antitumor immunity induced by SHP-2 or PD-1 targeting in myeloid cells.

## Discussion

Here we showed that SHP-2 and the PD-1-SHP-2 axis regulated myeloid cell differentiation and fate commitment and function in cancer. Myeloid-specific SHP-2 or PD-1 ablation-induced myeloid cells with enriched gene expression profiles of enhanced differentiation, activation, phagocytosis and features of effector differentiation. Monocytes from *Shp2^f/f^LysM^Cre^* and *Pdcd1^f/f^LysM^Cre^* tumor-bearing mice had a direct impact on controlling tumor growth and could transfer antitumor immunity into naïve hosts.

SHP-2 and the PD-1-SHP-2 signaling restrained the GM-CSF-mediated phosphorylation of HOXA10 and IRF8, which induce myeloid differentiation and monocyte/DC lineage commitment, respectively. During GM-CSF signaling, PD-1 interacted with Lyn and was phosphorylated at Y248 within the conserved ITSM motif, a site that is phosphorylated by Src family kinases in T cells leading to PD-1 interaction with SHP-2 (ref. [34]), indicating canonical PD-1-SHP-2 signaling axis was operative in myeloid cells, similarly to what has been previously established for B and T lymphocytes[32]. This myeloid-specific PD-1-SHP-2 axis might be particularly important in the context of cancer, where growth factors released by cancer cells induce emergency myelopoiesis, and directly upregulate PD-1 and PD-L1 expression in myeloid progenitors and their progeny, thereby posing a signaling restrain to their effector differentiation. This might be the earliest critical signaling target of the PD-1-PD-L1 pathway in the context of cancer, because soluble factors produced by cancer cells can act systemically at early stages of the cancer immunity cycle, before local tumor growth and infiltration by immune cells that are subject to inhibitory signals in the tumor microenvironment.

We found that targeting SHP-2 in myeloid cells resulted in enhanced differentiation of neutrophils and TAMs generated from bone marrow-derived monocytes in tumors. Both cell types were characterized by enriched gene signatures and pathways linked to effector differentiation, leukocyte-mediated immunity, cytokine and chemokine production. TAM-mediated chemokine production and chemotaxis have been reported to have a tumor-promoting effect by recruiting immunosuppressive myeloid cells to tumors[47]. In contrast, chemokine production signatures of TAMs in *Shp2^f/f^LysM^Cre^* and *Pdcd1^f/f^LysM^Cre^* tumor-bearing mice were associated with diminished tumor growth. This outcome might be explained by the differentiated state of SHP-2-deficient and PD-1-deficient myelocytes that were skewed away from protumorigenic MDSCs and TAMs and, instead, had enhanced effector functions promoting antitumor immune responses.

The impact of inhibiting SHP-2 in myeloid cells has been previously reported by using allosteric inhibitors of SHP-2 such as SHP099, TNO155 (ref. [48]) and RMC4550 (ref. [49]) to target cancer cells where SHP-2 is activated downstream of RTK/Ras signaling and functions as an oncogene. Combined approaches of SHP-2 allosteric inhibitors and pharmacologic RTK inhibitors or KRAS^G12C^-GDP inhibitors have been employed with the main purpose to target signaling vulnerabilities in cancer[50]. Such treatments also altered immune cells of the TME but remained elusive whether this was due to a direct effect of these compounds on immune cells or a consequence of inhibiting cancer growth. Our studies, employing a genetic approach, showed an increase of differentiated granulocytes and TAMs with signatures of effector differentiation and leukocyte-mediated immunity indicating a direct effect in myeloid cells after SHP-2 ablation.

IL-10, an IRF8-regulated gene[45], was increased in SHP-2-deficient and PD-1-deficient TAMs, whereas IL-10 neutralization compromised the enhanced antitumor immunity of *Shp2^f/f^LysM^Cre^* and *Pdcd1^f/f^LysM-^Cre^* mice. Myeloid-specific SHP-2 ablation was reported to enhance IL-10-mediated immunosuppression in gut macrophages protecting mice from intestinal inflammation and carcinogenesis[51]. In our system, we did not find detectable effects of IL-10 in the immunosuppressive function of MDSCs or the activation state of TAMs. IL-10 is currently emerging as a previously unappreciated regulator of antitumor function and a master switch of tumor-promoting inflammation to antitumor immunity[46]. This might be mediated by IL-10-receptor-mediated recruitment and engagement of STAT3, diminishing procarcinogenic IL-6-mediated STAT3 signaling. IL-10 might have direct effects on T cells because pegylated IL-10 induced systemic activation and polyclonal expansion of CD8^+^ T cells in cancer patients[52].

Besides IL-10, TAMs from *Shp2^f/f^LysM^Cre^* and *Pdcd1^f/f^LysM^Cre^* mice had increased transcripts of other genes that counteract protumorigenic inflammation, compared to *Shp2^f/f^* and *Pdcd1^f/f^* TAMs. For instance, there was an increase in *IRAK3*, which functions as a negative regulator of MyD88-mediated proinflammatory activation downstream of TLR/IL-1R[53]. We also found a common increase in the expression of *Zbtb18*, a zinc finger transcriptional repressor recently identified to bind promoter/enhancer elements of genes encoding class I PI3K regulatory subunits, limiting their expression[54]. Because TLR/IL-1R and PI3K-mediated signaling in myeloid cells promote protumorigenic inflammation[55], our findings uncovered several mechanisms capable of mediating fine-tuning of proinflammatory signaling in SHP-2 and PD-1 deficient TAMs.

IRF8 has a decisive role in the differentiation of monocytes and DC via epigenetic regulation of distinct sets of enhancers in cooperation with other transcription factors. High, low and null IRF8 expression promotes the differentiation of conventional DC, monocytes and neutrophils, respectively[45]. In the absence of SHP-2- or PD-1-mediated signals, bone marrow myeloid cells had enhanced IRF8 phosphorylation, which is required for IRF8 function[13]. IRF8 expression is fine-tuned at various differentiation stages of the myeloid lineage. Its expression starts at the multipotent progenitor stage, substantially increases in MDP and further increases in CDP and cells differentiating into the DC lineage, whereas it remains relatively low or downregulated in the monocytic lineage[45]. In addition to transcription, IRF8 abundance is regulated by post-transcriptional mechanisms through Notch[56] and c-Cbl ubiquitin ligase[57]. Future studies are required to determine how IRF8 abundance and function are regulated in the context of cancer and PD-1-SHP-2 signaling.

Our results showed that monocytes isolated from the bone marrow of SHP-2 and PD-1 deficient tumor-bearing mice could transfer antitumor capacity to naïve hosts, a feature previously attributed to the development of trained immunity. Neutrophils might also develop anticancer-protective trained immunity[30]; however, in our system, we found that monocytes but not neutrophils conferred antitumor protection. Of note, IL-1β that drives training of monocyte precursors[31] was highly increased in SHP-2 and PD-1 deficient TAMs, potentially explaining the preferential development of training in monocytes by blockade of the PD-1-SHP-2 signaling. It is tempting to speculate that the long-lasting effects of PD-1-based immunotherapy in some, but not all, patients might be related to the development of immunotherapy-mediated monocyte differentiation and trained immunity versus generation of immunosuppressive MDSCs and TAMs during cancer-driven emergency myelopoiesis. Future studies will investigate these new directions of central regulation of antitumor responses to cancer immunotherapy in patients.

In conclusion, our results provide multiple levels of evidence that SHP-2 and the PD-1-SHP-2 axis pose a signaling restrain to the differentiation and monocyte/DC lineage in the context of cancer, resulting in a myeloid landscape that compromises antitumor immunity.

## Online content

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

## Methods

### Mice

All mice procedures were approved by the Institutional Animal Care and Use Committee at Beth Israel Deaconess Medical Center (Boston MA) and were in accordance with the National Institutes of Health Guidelines for the Care and Use of Animals. An approved active protocol is in place for the investigator (046-2019). The studies are compliant with the maximum tumor size permitted by the committee and tumors were never allowed to ulcerate. Mice with conditional targeting of the *Ptpn11* gene (encoding for Shp-2) were kindly provided by Dr. Gen-Sheng Feng (University of California, San Diego) and have been previously described[58]. Mice with conditional targeting of the *Pdcd1* gene (encoding for PD-1) have been previously described[7]. Mice expressing Cre recombinase under the control of the distal Lck promoter (Strain, 012837) and mice expressing Cre recombinase under the control of the lysozyme (LysM) promoter (Strain, 004781), C57BL/6 mice and OTI-TCR transgenic mice (C57BL/6-Tg(TcraTcrb)1100Mjb/J) were purchased from Jackson Laboratory (Bar Harbor, Maine).

### Tumor cell lines and tumor experiments

B16-F10 melanoma and MC17-51 fibrosarcoma cell lines were purchased from ATCC. B16-F10 cell line was subcloned and subclones with intermediate growth rate were selected for use. Eight to twelve weeks old mice were used for tumor implantation, and B16-F10 melanoma ($1 \times 10^5$ or $3 \times 10^5$ cells per mouse, as described in individual experiments) or MC17-51 fibrosarcoma ($4 \times 10^4$ or $1 \times 10^5$ cells per mouse) were injected subcutaneously in the left flank under isoflurane anesthesia. For adoptive transfer of monocytes or neutrophils into naïve hosts, CD45$^+$CD11b$^+$Ly6C$^{hi}$Ly6G$^-$ monocytes and CD45$^+$CD11b$^+$Ly6C$^-$Ly6G$^+$ neutrophils were magnetically collected from the bone marrow of tumor-bearing mice 9 days after implantation of MC17-51 tumors. Further, $0.75 \times 10^5$ monocytes or $0.75 \times 10^5$ neutrophils were mixed with equal number of MC17-51 cells and were injected subcutaneously in new WT C57BL/6 hosts. Starting from day 9, tumor size was monitored every 2 days by a caliper fitted with a Vernier scale. The tumor volume was calculated by the following formula: tumor volume = $0.5 \times \text{width}^2 \times \text{length}$. Mice were killed on days 14–16 after tumor implantation and bone marrows, spleens, tumors and tumor-draining lymph nodes were collected. At termination, tumor weight was also measured. For antibody treatment experiments, 250 μg of either InVivoMAb anti-PD-1 (clone RMP1-14, BioXCELL) or IgG2a control (clone 2A3, BioXCell) diluted in sterile PBS were injected intraperitoneally in a volume of 100 μl per mice on days 9, 11 and 13 after tumor inoculation and the mice were euthanized at day 15. For T cell depletion, mice were injected on day 1 relatively to tumor inoculation, and subsequently every third day with 200 μg of InVivoMAb antimouse CD3ε (clone 145-2C11 f(ab')2 fragments, Bio X Cell) or hamster IgG f(ab')2 fragments (BioXCell). For treatment with anti-IL-10 antibody, mice were treated with 250 μg of either InVivoMAb anti-IL-10 (clone JES5-2A5, BioXCell) or rat IgG1 control (clone HRPN, BioXCell) on days 9, 11 and 13 after tumor inoculation.

### Cell purification and processing

Single-cell suspensions were made from spleens, tumor-draining lymph nodes, and tumors as previously described[7]. Briefly, tumors were digested by 1 mg ml$^{-1}$ of Collagenase I in incomplete RPMI and then filtered through 70 μl strainers, while the spleens and the tumor-draining lymph nodes were directly filtered through 70 μl strainers. For flow cytometry studies, $1 \times 10^6$ were resuspended in 1X PBS supplemented with 2.5% FBS and plated in 96-well round bottom plates. Surface staining was performed at 4 °C for 25–30 min with the flow antibodies listed in Supplementary Table 7. For intracellular staining, Foxp3/transcription factor permeabilization/staining Buffer set (Thermo Fischer Scientific) was used according to the manufacturer's instructions. Intracellular staining was performed at 4 °C for 35–45 min with flow antibodies listed in Supplementary Table 3. Cells were acquired

using Becton Dickinson LSR Fortsessa or Beckman-Coulter Cytoflex flow cytometer and analyzed with FlowJo software.

### Suppression assay

MDSC-mediated suppression was assessed using previously established methodology[7]. Briefly, splenic MDSCs were isolated from the spleens of mice bearing B16-F10 melanoma or MC17-51 fibrosarcoma, by using the EasySep Mouse (CD11b$^+$GR1$^+$) isolation kit (Stemcell Technologies, 19867), or the MDSC isolation kit (Miltenyi Biotech, 130-094-538) to separate GR1$^{hi}$Ly6G$^+$Ly6C$^-$ (PMN-MDSC) and GR1$^{dim}$Ly6G$^-$Ly6C$^+$ (M-MDSC) cells. Serial dilutions of MDSCs ($2 \times 10^5$, $1 \times 10^5$, $0.5 \times 10^5$, $0.25 \times 10^5$ and $0.125 \times 10^5$) were plated in flat bottom 96-well plates with $2 \times 10^5$ splenocytes per well isolated from OTI-TCR transgenic mice and 250 ng ml$^{-1}$ of ovalbumin peptide (OVA$_{257-264}$) for 72 h. As a control, OTI splenocytes were incubated with OVA peptide (OVA$_{257-264}$) without MDSC. $^3$H-thymidine was added for the last 16 h of a 72 h culture, and thymidine incorporation was measured by MicroBeta plate counter (Perkin Elmer).

### Cell culture and signaling

For signaling experiments, bone marrow cells from C57BL/6 WT mice or *Shp2$^{f/f}$* and *Shp2$^{f/f}$LysM$^{Cre}$* mice were cultured for 48 h in Iscove's media supplemented with 10% FBS, 2 mmol l$^{-1}$ glutamine, 100 units per ml penicillin–streptomycin, 10 mM Hepes and 20 μM β-mercaptoethanol, in the presence of GM-CSF (10 ng ml$^{-1}$) and IL-3 (5 ng ml$^{-1}$) (both purchased from Peprotech). Where indicated, a PD-L1 blocking antibody (MIH5) (10 μg ml$^{-1}$) was added to the cultures for the entire period of incubation. For studies of GM-CSF-mediated short-term signaling activation, 48 h after culture as described above, cells were rested for 3 h at 37 °C in RPMI 1640 containing 10 mM HEPES. Cells were then either left unstimulated or resuspended at $10 \times 10^6$ cells per ml in prewarmed RPMI 1640 containing 10 mM HEPES and were stimulated with GM-CSF (40 ng ml$^{-1}$). At the indicated time points, the reaction was stopped by adding cold PBS and placement on ice.

### Immunoprecipitation and immunoblotting

To prepare lysates, cells were washed in PBS and lysed as previously described in ref. 34. Briefly, cells were resuspended and lysed in lysis buffer containing 50 mM Tris–HCl, pH 7.4, 150 mM NaCl, 2 mM MgCl$_2$, 10% glycerol and 1% NP-40 supplemented with 2 mM sodium orthovanadate, 1 mM sodium fluoride, 1 mM phenylmethylsulfonyl fluoride and protease Inhibitor Cocktail (Thermo Fischer Scientific). Cell lysates were resolved by SDS-PAGE, transferred on nitrocellulose membrane, and analyzed by western blotting with the indicated antibodies. The following antibodies were used for western blotting: SHP-2 (D50F2) 3397T, Cell Signaling Technology; Lyn (H-6) sc-7274 AF790, Santa Cruz Biotechnology; HoxA10 (E-11) sc-271428 AF680, Santa Cruz Biotechnology; ICSBP (IRF8) (E-9) sc-365042 AF680, Santa Cruz Biotechnology; Pdcd-1 (RMP1-30) sc-56200 AF680, Santa Cruz Biotechnology; IL-3/IL-5/GM-CSFRβ (A-3) sc-398246, Santa Cruz Biotechnology; p-Tyr (PY99) sc-7020 AF790, Santa Cruz Biotechnology. The rabbit polyclonal antiphospho-Y248 (ITSM) PD-1 (pPD-1) antibody was developed in our laboratory[34]. For conjugation of pPD-1 Ab, the Li-COR IRDye 800CW protein labeling kit for high molecular weight and microscale reactions (829-08881) was used. Immunoprecipitations were performed with agarose-conjugated antibodies HoxA10 (E-11) sc-271428 AC, Santa Cruz Biotechnology; ICSBP (IRF8) (E-9) sc-365042 AC, Santa Cruz Biotechnology; Pdcd-1 (RMP1-30) sc-56200 AC, Santa Cruz Biotechnology. Briefly, 20 μl of agarose slurry/sample was first washed three times with lysis buffer and then resuspended in 40 μl of buffer. For each IP sample, 40 μl of washed agarose-conjugated Ab were mixed with 500–1000 μg of cell lysates and incubated overnight at 4 °C with gentle rotation. The agarose slurry was then washed three times with lysis buffer and boiled for 5 min in denaturing sample buffer followed by a quick spin. The supernatant was analyzed by SDS-PAGE, transferred to

a nitrocellulose membrane and blotted with the indicated antibodies. Images were visualized, acquired and quantified with Li-COR Odyssey CLx imaging system. The abundance of phosphorylated HOXA10 and IRF8 was normalized to the immunoprecipitated HOXA10 and IRF8, respectively, and was expressed as fold change over the value obtained in *Shp2^{f/f}* cells, defined as one. Expression of actin in whole-cell lysates was used as input. The abundance of SHP-2 coprecipitated with PD-1 was normalized to the immunoprecipitated PD-1 and was expressed as a fold change over the value obtained in *Shp2^{f/f}* cells, defined as one. Expression of PD-1 in whole-cell lysates was used as input.

### RNA-seq and analysis

For RNA-seq, PMN-MDSCs (GR1^{hi}Ly6G^+Ly6C^−) were isolated from the spleens of MC17-51 fibrosarcoma bearing and *Shp2^{f/f}LysM^{Cre}* and *Shp2^{f/f}* mice by magnetic bead isolation. TAMs were isolated from the same mice by cell sorting after using the Live/Dead Fixable Far Read Dead Cell Stain kit (Thermo Fischer Scientific; L34973) to identify live cells, staining with antibodies specific for CD45, CD11b, F4/80 and Ly6G and gating on CD45^+CD11b^+F4/80^+Ly6G^− live cells. Total RNA was isolated from the cells using the Qiagen RNeasy Mini Kit (Qiagen, 74104). For each sample, 400 ng of total RNA was then used in Illumina's TruSeq Stranded mRNA Library kit (20020594) for polyA mRNA isolation and library construction. Libraries were sequenced on Illumina NextSeq 500 as paired-end 42-nt reads (Active Motif). Raw sequencing reads were quality-checked using FastQC (v0.11.5)[59] and data were pre-processed with Cutadapt (v2.5)[60] for adapter removal following best practices[61]. Gene expression quantification was performed by aligning against the GRCm38 genome using STAR (v2.7.3a)[62] and quantifying reads against Ensembl v98 (ref. 63) annotated gene loci with feature-Counts (Subread 1.6.2)[64]. Differential gene expression analysis was performed using DESeq2 (v1.24.0)[65], while ClusterProfiler (v3.12.0)[66] was used for downstream functional investigations. Plots were generated in R using ggplot2 (v3.3.3)[67], EnhancedVolcano (v1.8.0)[68] and ComplexHeatmap (v2.6.2)[69]. Storey's *q* value was used to control family-wise error rate[70]. Sequencing data have been deposited in the Gene Expression Omnibus database under the accession numbers GSE187394 and GSE206207.

### Metabolite analysis

Phagocytes were differentiated from bone marrow of *Shp2^{f/f}* and *Shp2^{f/f}LysM^{Cre}* mice by culture in Iscove's media supplemented with 10% fetal bovine serum, 2 mmol l^{−1} glutamine, 100 units per ml penicillin–streptomycin, 10 mM Hepes and 20 μM beta-mercaptoethanol, in the presence of GM-CSF (40 ng ml^{−1})[71]. Polar metabolites were quantitatively profiled by a positive/negative ion-switching, targeted liquid chromatography–tandem mass spectrometry (LC–MS/MS) based metabolomics platform using a 5500 QTRAP hybrid triple quadruple mass spectrometer (AB/SCIEX) via selected reaction monitoring (SRM) as described previously[72]. Briefly, Lin^{neg} bone marrow cells were cultured with G-CSF and GM-CSF (40 ng ml^{−1} each) for 48 h using triplicate samples for each condition and sample type. After methanol extraction using 80% (vol/vol) methanol (−80 °C) was carried out, pellets were lyophilized using a SpeedVac concentrator using no heat. Twenty microliter of LC–MS grade water was added to resuspend each sample just before LC–MS/MS analysis and 5 μl of sample was injected onto the autosampler of the LC system (Shimadzu) using an amide HILIC column (Waters). Once the SRM data for ~285 metabolites were acquired, peaks were integrated using a software platform for peak area integration MultiQuant 2.1 (AB/SCIEX). Data analysis was performed using online MetaboAnalyst 3.0 software.

### Real-time quantitative PCR

Total RNA extraction was prepared with the RNeasy Mini Kit from Qiagen, Valencia, CA, according to the manufacturer's instructions, and 50 ng of RNA was subjected to quantitative PCR (qPCR) analysis for the target genes *Clec4n* and *Cxcl3* using AB 7,000 qPCR machine (Applied Biosystems Roche). FAM-conjugated gene-specific primers for target genes and the TaqMan One-Step RT-PCR Master Mix reagents and the VIC-TAMRA-conjugated 18S RNA housekeeping gene control primers were from Applied Biosystems/Roche.

### Statistics and reproducibility

RNA-seq statistical analysis was completed as described above. All other statistical analyses were performed using GraphPad Prism (GraphPad Software v.9.3.0). Values are given as mean ± s.d. as indicated. Numbers of experimental replicates are given in the figure legends. When two groups were compared, significance was determined using an unpaired two-tailed *t*-test. For comparing more than two groups, one-way analysis of variance was applied. $P < 0.05$ are considered as statistically significant (*$P < 0.05$, **$P < 0.01$, ***$P < 0.001$, ****$P < 0.0001$). No statistical methods were used to predetermine sample sizes, and our sample sizes were similar to those reported in the previous publications[7,16,19]. Mice were assigned randomly to the various experimental groups described. Equal numbers of male and female mice were used in all experiments. Data collection and analysis were not performed blind to the conditions of the experiments. Data distribution was assumed to be normal, but this was not formally tested. No data points were excluded from the analyses.

### Reporting summary

Further information on research design is available in the Nature Portfolio Reporting Summary linked to this article.

## Data availability

All data generated or analyzed during this study are included in this published article (and its supplementary information files). Sequencing data have been deposited in the Gene Expression Omnibus database under the accession numbers GSE187394 and GSE206207 and are publicly available. Source data are provided with this paper.

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

## Acknowledgements

This work was supported by NIH/NCI under grants CA238263 and CA229784 (to V.A.B.).

## Author contributions

A.C. performed most experiments, wrote sections of the manuscript and prepared figures; C.C., K.A., S.Y., I.A., M.A.A.M., L.S., N.M.T. and R.P. performed experiments; X.L.K., D.K. and I.S.V. performed bioinformatics analysis and prepared figures; L.B. provided reagents; J.A. did metabolomics; N.P. did experiments, participated in the experimental design and wrote sections of the manuscript and V.A.B. designed the project, guided participants, did experiments and wrote the manuscript. All authors read, edited and approved the manuscript.

## Competing interests

V.A.B. has patents on the PD-1 pathway licensed by Bristol-Myers Squibb, Roche, Merck, EMD-Serono, Boehringer Ingelheim, AstraZeneca, Novartis and Dako. All the other authors declare no competing interests.

## Additional information

**Extended data** is available for this paper at https://doi.org/10.1038/s41590-022-01385-x.

**Correspondence and requests for materials** should be addressed to Vassiliki A. Boussiotis.

**Peer review information** *Nature Immunology* thanks Cecilia Garlanda and the other, anonymous, reviewer(s) for their contribution to the peer review of this work. (if applicable to your journal): Ioana Visan was the primary editor of this article and managed its editorial process and peer review in collaboration with the rest of the editorial team. Peer reviewer reports are available.

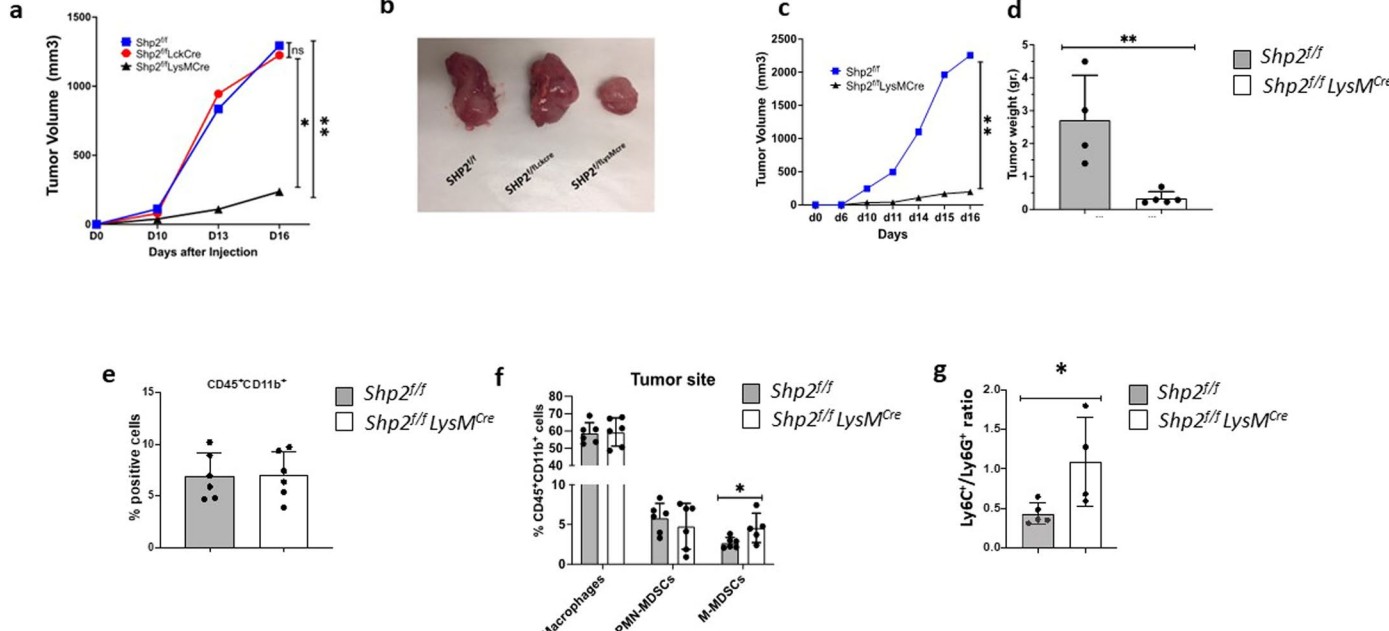

**Extended Data Fig. 1 | Myeloid-specific SHP-2 depletion diminishes tumor growth. a**, *Shp2$^{f/f}$*, *Shp2$^{f/f}$Lck$^{Cre}$*, and *Shp2$^{f/f}$LysM$^{Cre}$* mice were inoculated with MC17-51 fibrosarcoma cells, tumor volume was monitored longitudinally, and comparisons were made on each day of assessment. Data shown are means of n = 6 mice per group and are representative of three independent experiments. (*p < 0.05, **p < 0.01), ANOVA. **b**, Representative images of tumors isolated at day 15 from each of the three experimental group are shown. **c, d**, *Shp2$^{f/f}$* and *Shp2$^{f/f}$LysM$^{Cre}$* mice were inoculated with B16-F10 melanoma cells, tumor volume was monitored longitudinally (c) and tumor weight was measured at termination on day 16 (d). Data are means of n = 5 mice per group and are representative from one of four independent experiments. (*p < 0.05, **p < 0.01), unpaired t-test two tailed. **e-g**, The frequencies of CD45$^+$CD11b$^+$ myeloid cells (e), the fractions of macrophages, PMN-MDSC and M-MDSC (f) and the ratio of M-MDSC/PMN-MDSC (g) in tumors were assessed. Data are representative of means ± SD are shown. Results from one representative of 4 independent experiments with n = 4 mice per group are shown (*p < 0.05, **p < 0.01), unpaired t-test two tailed.

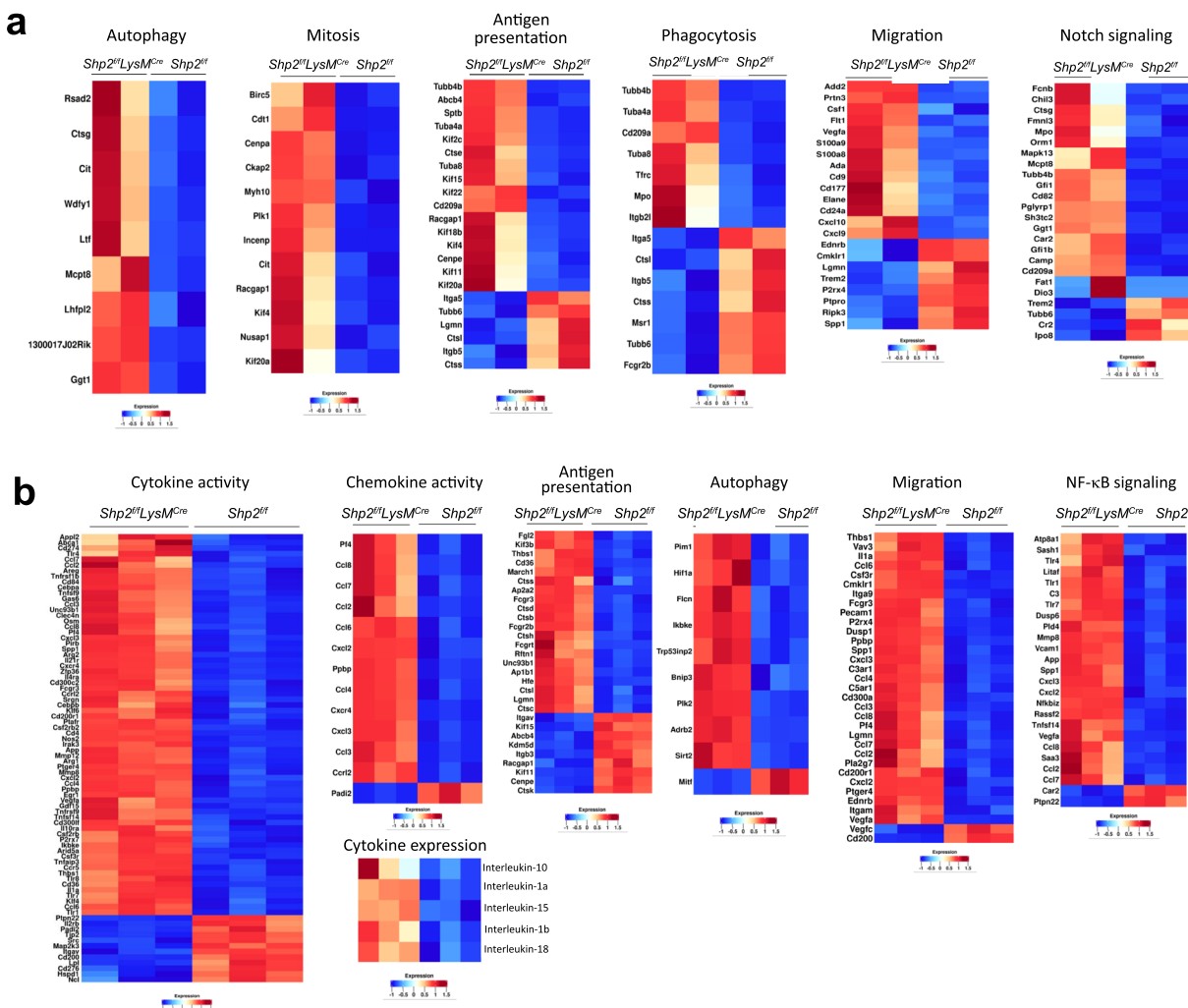

**Extended Data Fig. 2 | Distinct transcription signatures in PMN-MDSC and TAMs of *Shp2f/fLysMCre* tumor-bearing mice compared to *Shp2f/f* tumor-bearing mice. a, b**, *Shp2f/fLysM^Cre* and *Shp2f/f* mice were injected with MC17-51 cancer cells and 15 days later, PMN-MDSC (a) were isolated from the spleens,

TAMs (b) were isolated from tumors, and RNA-seq was performed followed by pathway enrichment analysis of DEG. Heat maps of DEG for enriched pathways are shown. Differential gene expression analysis was performed using DESeq2.

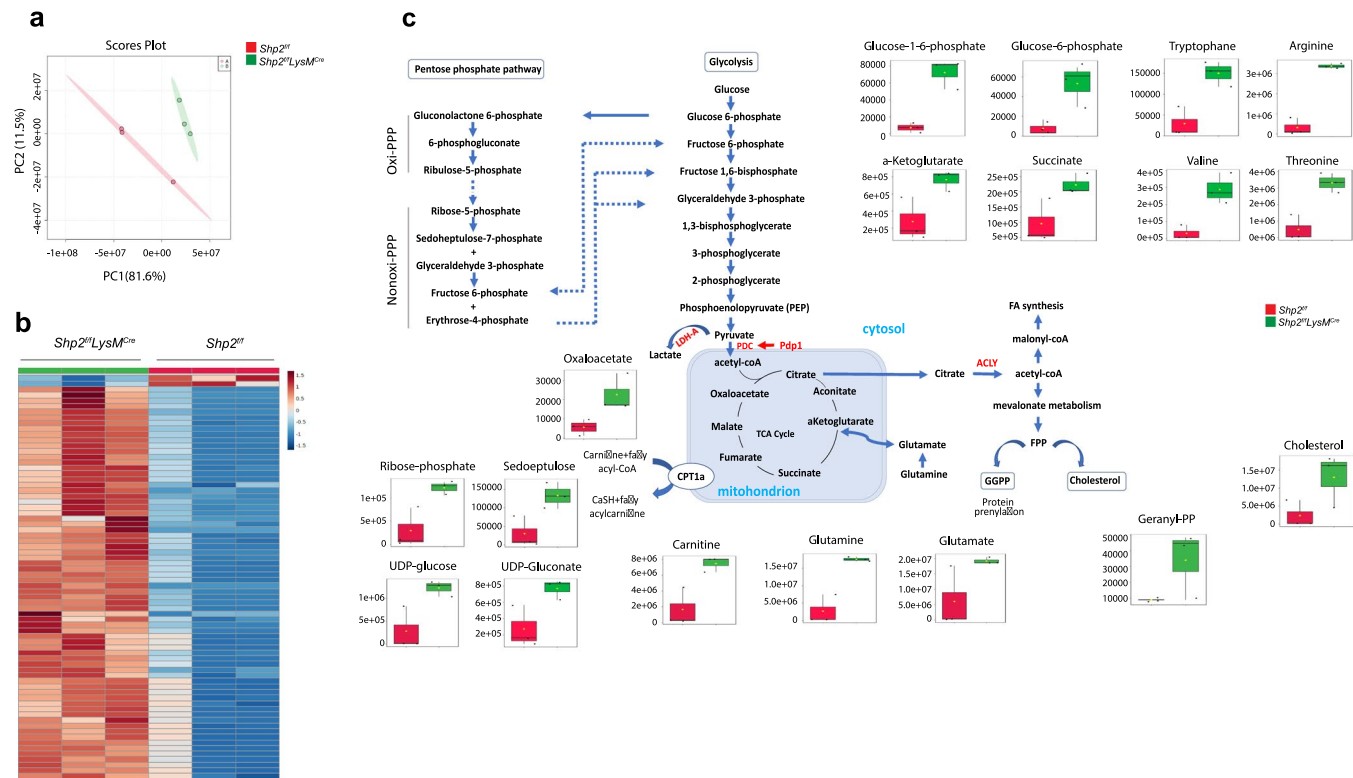

**Extended Data Fig. 3 | Phagocytes from *Shp2f/fLysMCre* and *Shp2f/f* mice have distinct metabolic activities. a**, Phagocytes were generated from primary bone marrow cells of *Shp2f/fLysMCre* and *Shp2f/f* mice using GM-CSF and metabolite analysis was performed after 48 hours of culture. Principal component analysis (PCA). **b**, Unsupervised hierarchical clustering of top 75 metabolites (log2FC ≥ 1). **c**, Individual graphs of relative peak intensity of representative intermediate metabolites of glycolysis, PPP and TCA cycle. Results from one representative of two independent experiments are shown. The amounts of the indicated metabolites were plotted in whisker boxes. The lower and upper sides of the box indicate the first and third quartile, respectively. The horizontal line inside the box indicates the median value, whereas the lower and upper bars indicate the minimum and maximum of distribution, respectively.

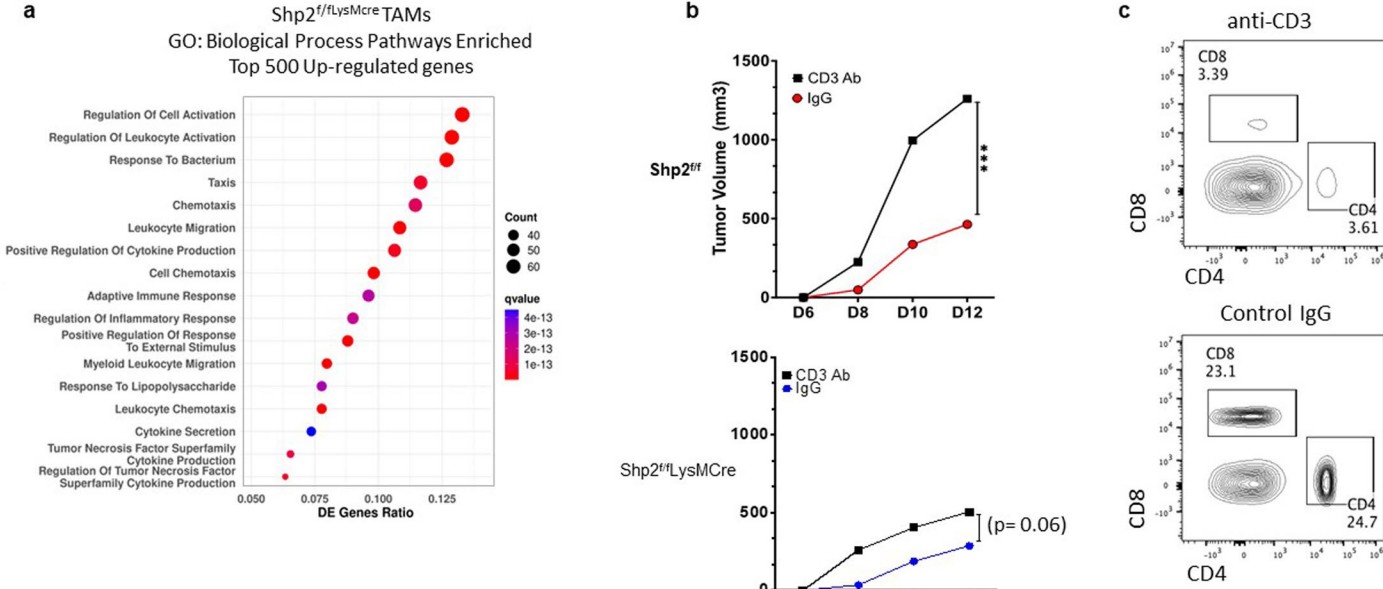

**Extended Data Fig. 4 | Myeloid cells of *Shp2f/fLysMCre* mice have distinct molecular and functional properties. a**, GO Biological Processes Pathways enriched among top 500 upregulated genes in TAMs from *Shp2^{f/f}LysM^{Cre}* MC17-51 tumor-bearing mice compared to *Shp2^{f/f}* MC17-51 tumor-bearing mice, collected at day 15 after tumor implantation. Differential gene expression analysis was performed using DESeq2 and ClusterProfiler (v3.12.0) was utilized for downstream functional investigations. **b, c**, *Shp2^{f/f}* and *Shp2^{f/f}LysM^{Cre}* mice injected with MC17-51 fibrosarcoma were treated with either anti-CD3 antibody or control IgG at day -1 relative to tumor injection and subsequently every third day, and tumor growth was monitored for 12 days (b). Results show means of tumor volume and are representative of one from two independent experiments with n = 10 mice per group, (***p < 0.001) unpaired t-test two tailed. **c**, At termination, expression of lymph node CD4⁺ and CD8⁺ T cells was assessed by flow cytometry. One representative histoplot of each treatment condition generated from *Shp2^{f/f}LysM^{Cre}* mice is shown. (In this experiment, the number of injected MC17-51 cells was reduced by 50% because, after T cell depletion, tumors in *Shp2^{f/f}* mice rapidly exceeded the permitted size).

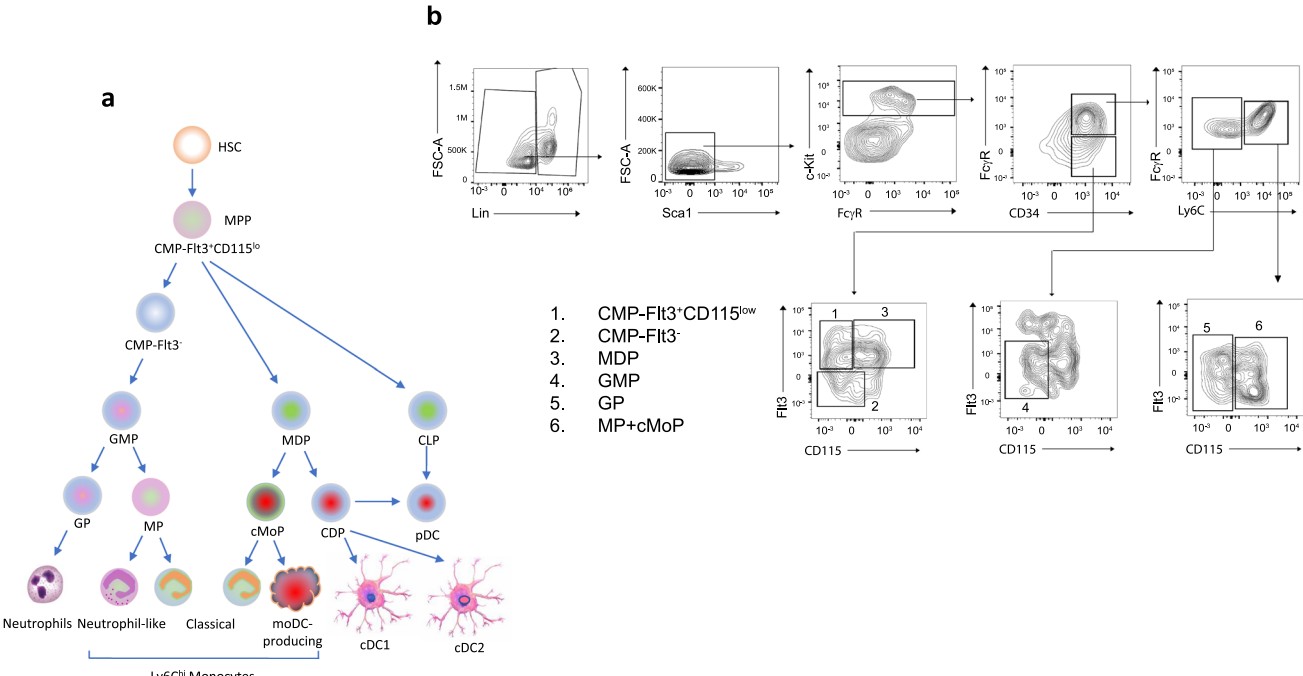

**Extended Data Fig. 5 | Differentiation and identification of myeloid progenitors. a, b**, Model for myeloid cell differentiation (**a**) and gating strategy (**b**) for characterization of bone marrow Lin⁻ myeloid progenitors. Identification of the progenitor subsets numbered in the histograms is shown.

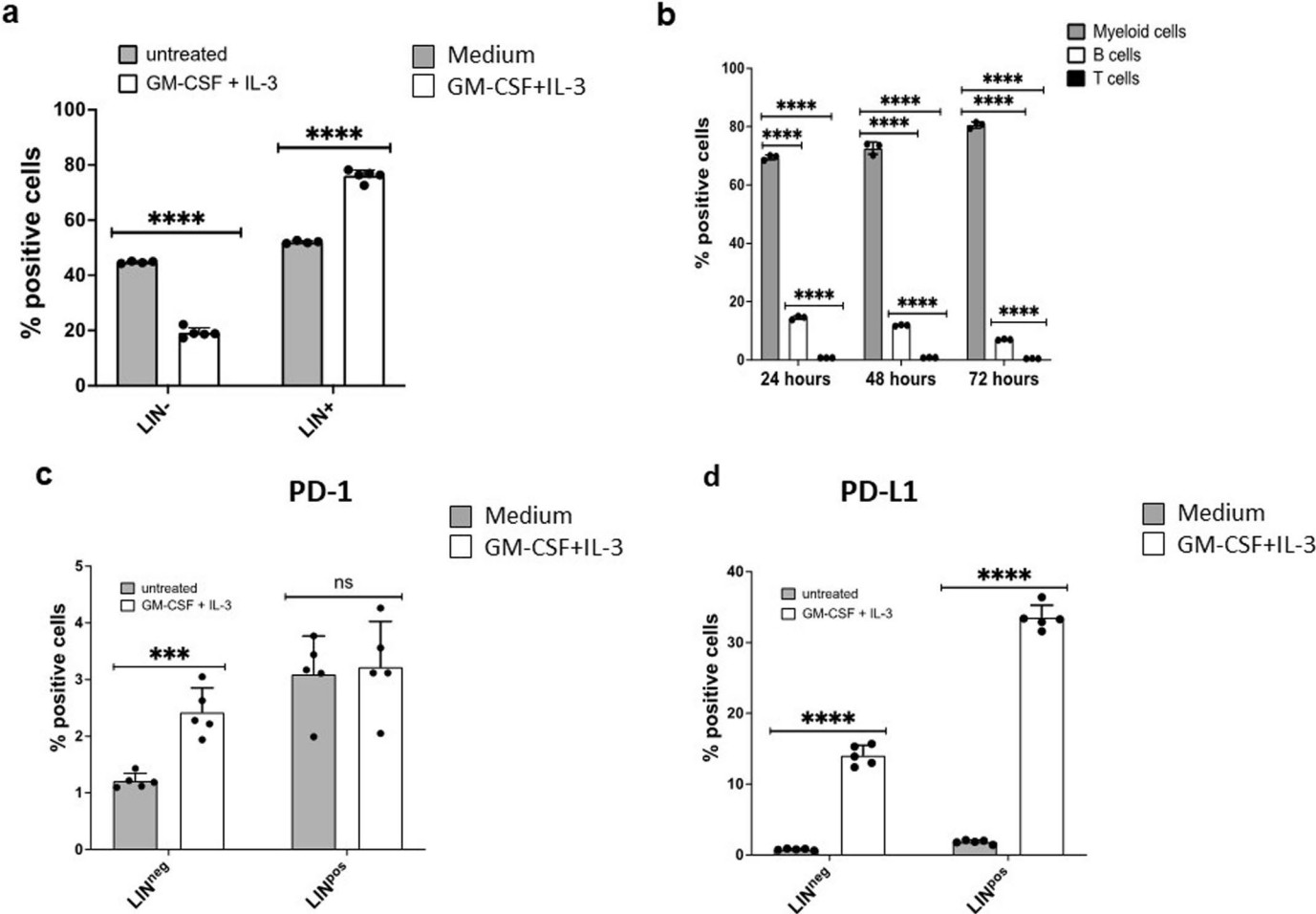

**Extended Data Fig. 6 | Culture of bone marrow cells with GM-CSF + IL-3 induces PD-1 and PD-L1 expression in Lin⁻ and Lin⁺ myelocytes. a b,** Bone marrow cells from WT C57BL/6 mice were cultured with GM-CSF (10 ng/ml) and IL-3 (5 ng/ml) for 24, 48 and 72 hours. Changes in the Lin⁻ and Lin⁺ populations were examined at 48 hours of culture (a). The frequency of the differentiated myeloid cells (CD45⁺CD11b⁺), B cells (B220⁺) and T cells (CD3⁺) was assessed by flow cytometry at the indicated time points (b). **c, d,** Expression of PD-1 (c) and PD-L1 (d) in Lin⁻ and Lin⁺ subsets examined by flow cytometry at 72 hours of culture. Results are from one of five independent experiments with n = 5 biological replicates per group (**p < 0.01, ***p < 0.005, ****p < 0.001) unpaired t-test two tailed.

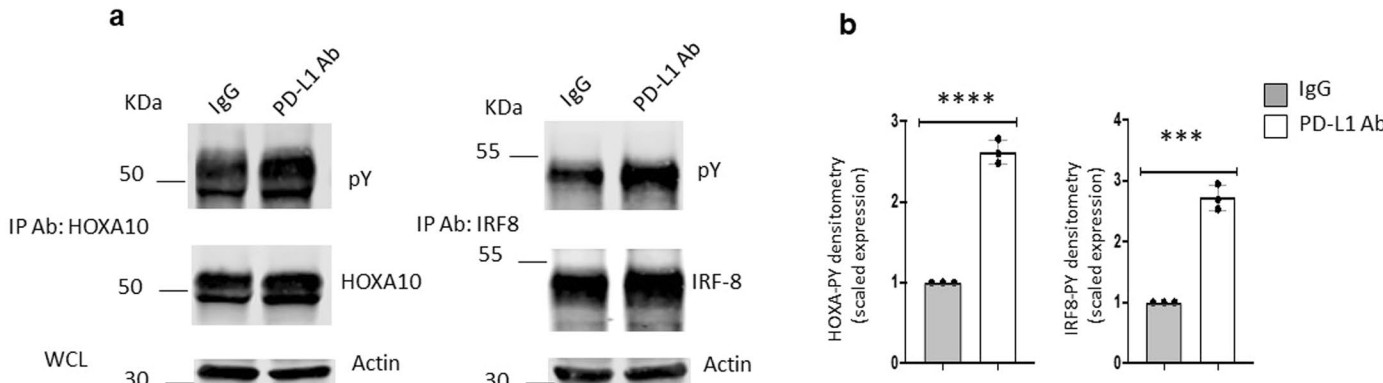

**Extended Data Fig. 7 | Enhanced HOXA10 and IRF8 phosphorylation in myeloid cells during culture with GM-CSF and IL-3 in the presence of anti-PD-L1 blocking antibody. a, b,** Bone marrow cells from C57BL/6 wild-type mice were cultured for 48 hr in the presence of GM-CSF (10 ng/ml) and IL-3 (5 ng/ml), with either IgG control of anti-PD-L1 blocking antibody (MIH5) (10 µg/ml). Cell lysates were prepared and immunoprecipitation was done with agarose-conjugated HOXA10-specific antibody or agarose-conjugated IRF8-specific antibody followed by SDS-PAGE and immunoblot with anti-PY and HOXA10 antibodies or anti-PY and and IRF8 antibodies (a). The abundance of phosphorylated HOXA10 was normalized to immunoprecipitated HOXA10 and the abundance of phosphorylated IRF8 was normalized to immunoprecipitated IRF8 and were expressed as fold change over the relevant values obtained in cells cultured without PD-L1 blocking antibody (defined as 1) (b). Expression of actin in whole cell lysates was also examined as input. Images were visualized, acquired and quantified with Li-COR Odyssey CLx imaging system. Results are from one of three independent experiments. Values of three separate quantifications per condition are shown.

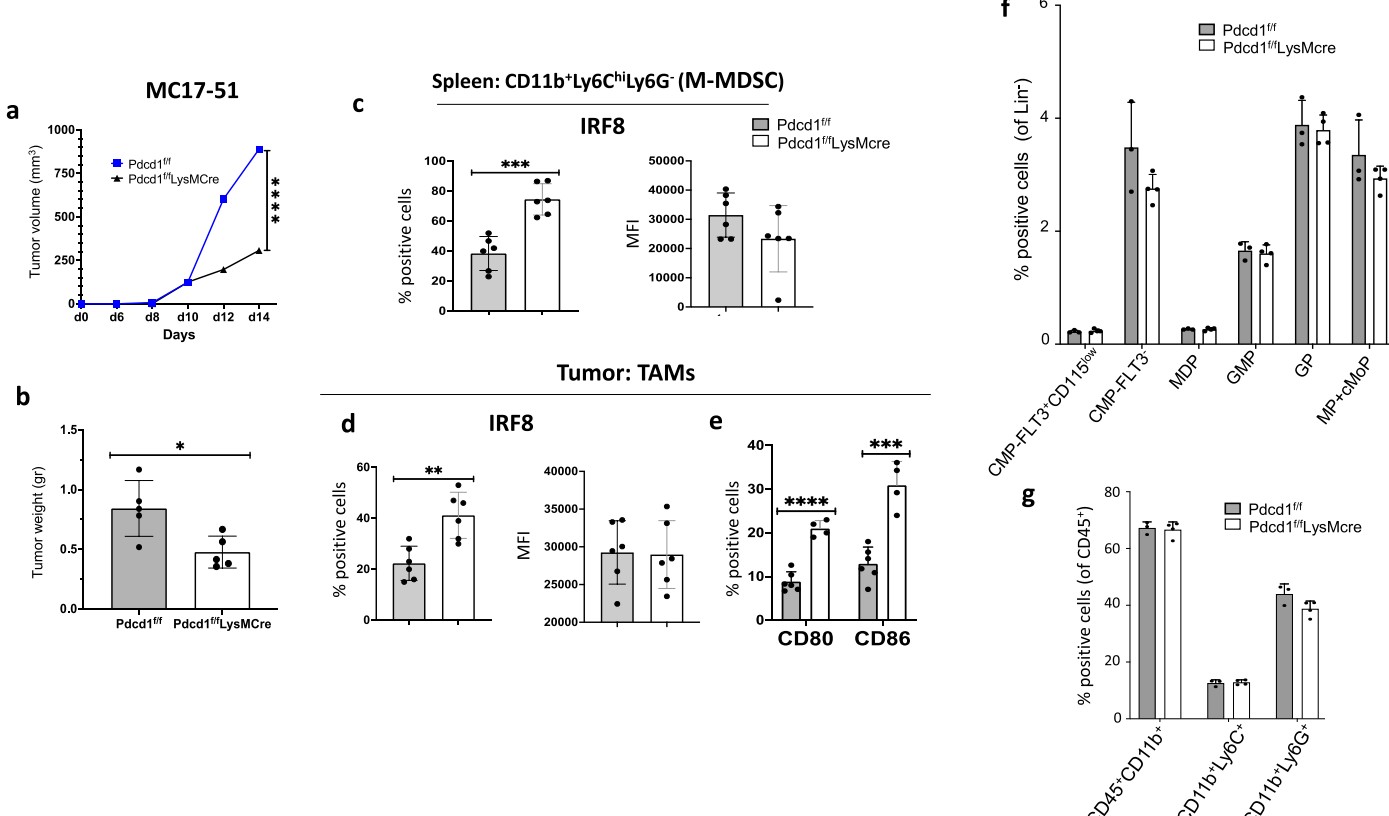

**Extended Data Fig. 8 | Myeloid-specific deletion of PD-1 induces antitumor immunity, and increased numbers of IRF8⁺ M-MDSC and TAMs, and CD80⁺ and CD86⁺ TAMs. a, b**, *Pdcd12^{f/f}* and *Pdcd1^{f/f}LysM^{Cre}* mice were inoculated with MC17-51 cells. Tumor volume was monitored longitudinally (a) and tumor weight (b) was measured at termination on day 15 after injection. Results show means of tumor volume (a) and means ± SD of tumor weight (b) and are representative of four independent experiments with n = 6 mice per group, (**p < 0.01, ****p < 0.0001), unpaired t-test two tailed. **c-e**, M-MDSC (c), and TAMs (d, e) were

examined for the expression of IRF8, CD80 and CD86 by flow cytometry. Means ± SD of % positive cells and MFI are shown. Results are from one representative of four independent experiments with n = 6 mice per group (**p < 0.01, ****p < 0.0001) unpaired t-test two tailed. **f, g**, At day 9 after tumor implantation, bone marrow was collected and flow cytometry was used to identify the subsets of Lin⁻ myeloid progenitors (f) using the gating strategy shown in Extended Data Fig. 5, the mature CD45⁺CD11b⁺ myeloid cells, and the subsets of Ly6C^{hi}Ly6G⁻ and Ly6C^{lo}Ly6G⁺ cells (g).

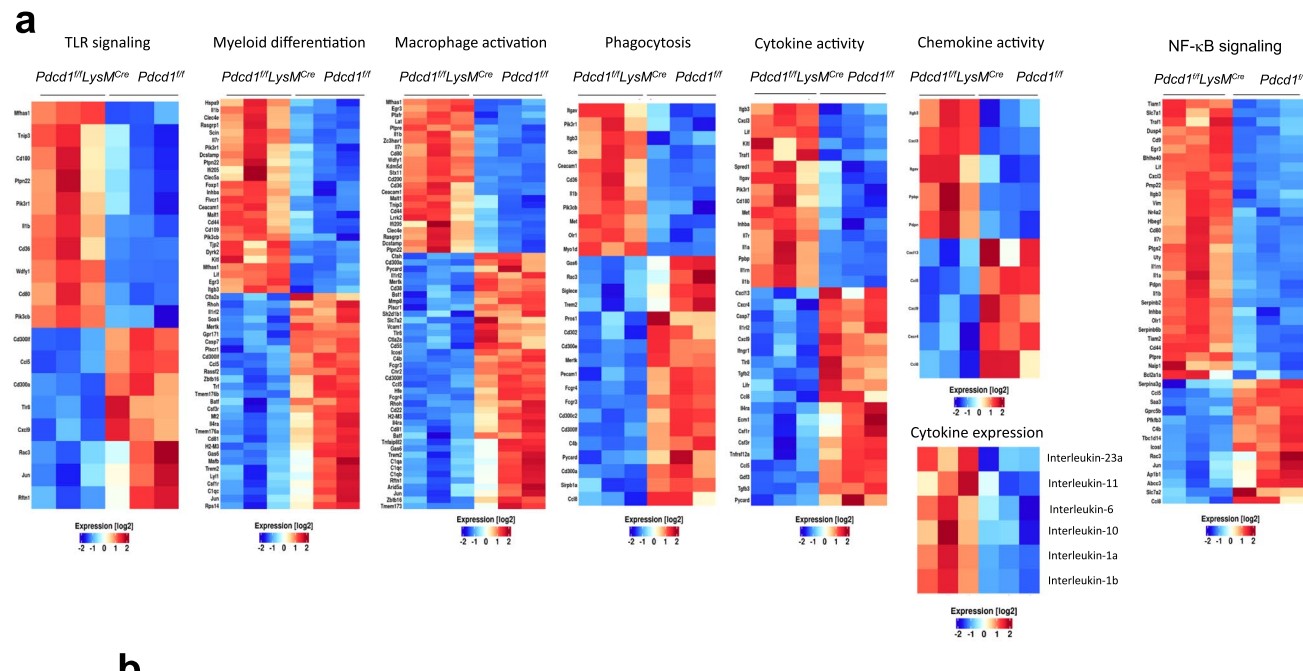

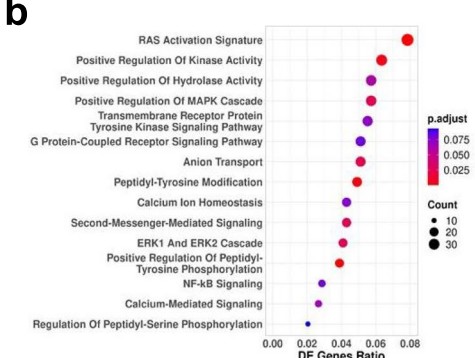

**Extended Data Fig. 9 | Distinct transcription signatures in TAMs of *Pdcd1f/fLysMCre* tumor-bearing mice compared to *Pdcd1f/f* tumor-bearing mice TAMs. a**, *Pdcd1f/fLysMCre* and *Pdcd1f/f* mice were injected with MC17-15 cancer cells and 15 days later, TAMs were isolated from tumors and RNA-seq was performed followed by pathway enrichment analysis of DEG. Heat maps of DEG for enriched pathways are shown. **b**, *Shp2f/fLysMCre* and *Pdcd1f/fLysMCre* mice were injected with MC17-51 fibrosarcoma and at day 15 after injection, TAMs were collected from tumors and RNA-seq was performed followed by GO analysis of DEG. GO Biological Processes of common signaling pathways enriched among top 500 DEG in TAMs from *Shp2f/fLysMCre* and *Pdcd1f/fLysMCre* tumor-bearing mice are shown. Differential gene expression analysis was performed using DESeq2 and ClusterProfiler (v3.12.0) was utilized for downstream functional investigations.

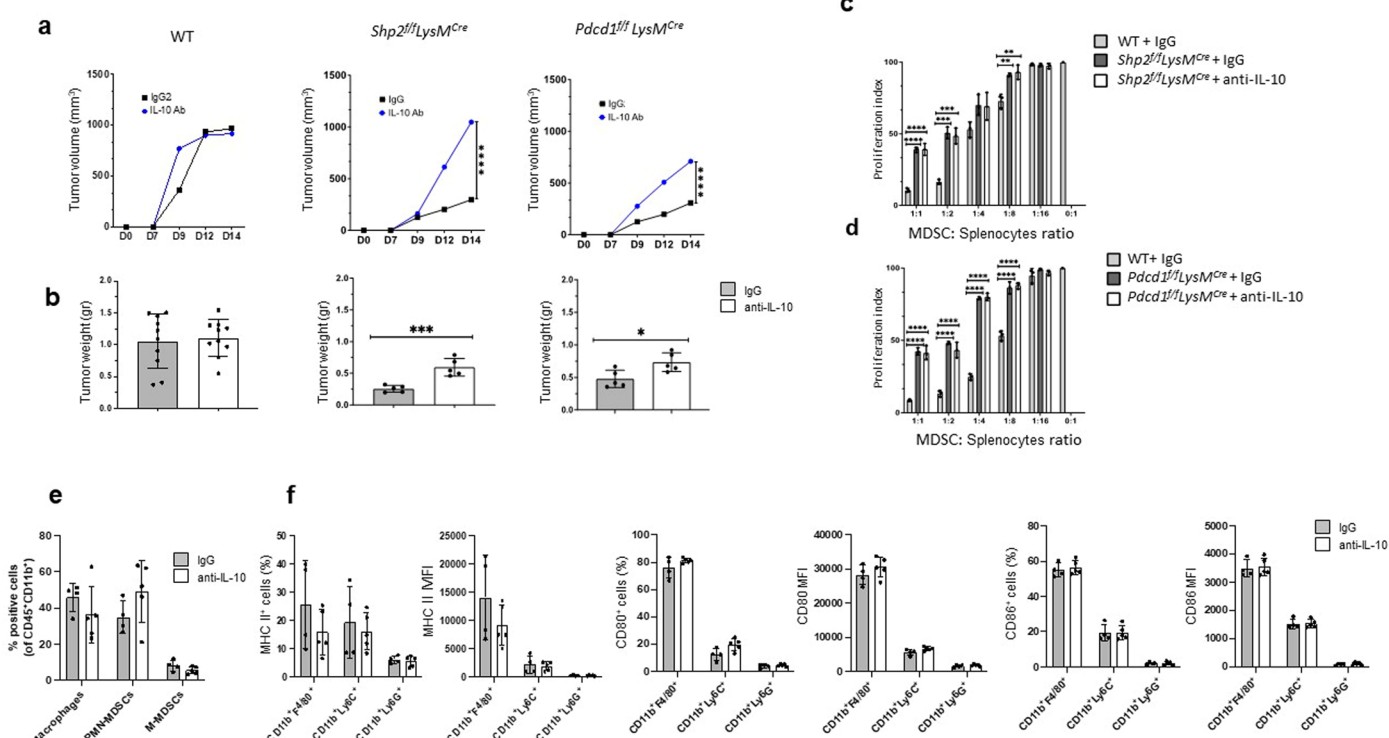

**Extended Data Fig. 10 | IL-10 neutralization compromises the enhanced anti-tumor responses of *Shp2f/fLysMCre* and *Pdcd1f/fLysMCre* mice. a-f,** Wild type, *Shp2f/fLysMCre* and *Pdcd1f/fLysMCre* mice were injected with MC17-51 cancer cells and were subsequently treated with anti-IL-10 Ab or control IgG1 on days 9, 11, and 13 after tumor inoculation. Tumor volume (a) was monitored longitudinally and tumor weight (b) was measured at termination on day 15 after injection. Results are representative of two separate experiments with n = 10 mice per group. At day 15 after tumor injection, GR1+MDSC were isolated from the spleens of wild-type mice treated with IgG, and from *Shp2f/fLysMCre* (c) and *Pdcd1f/fLysMCre* (d) tumor-bearing mice treated with anti-IL-10 Ab or IgG and were cultured at various ratios with splenocytes from OTI transgenic mice (2×10⁵cells/well) stimulated with OVA$_{257-264}$. Results show Means ± SEM of cpm values of ³H-thymidine incorporation and are representative of two separate experiments with n = 4 mice per group. At day 15 after tumor injection, the fractions of the indicated cell populations (e), and the expression of MHC II, CD80 and CD86 (f) at the tumor site, were examined by flow cytometry. Data show Means ± SD and are representative from one of two independent experiments with n = 5 *Shp2f/fLysMCre* mice per group (*p < 0.05, **p < 0.01, ***p < 0.001, ****p < 0.0001) unpaired t-test two tailed.

# Reporting Summary

## Statistics

For all statistical analyses, confirm that the following items are present in the figure legend, table legend, main text, or Methods section.

| n/a | Confirmed | |
|-----|-----------|---|
| ☐ | ☒ | The exact sample size (*n*) for each experimental group/condition, given as a discrete number and unit of measurement |
| ☐ | ☒ | A statement on whether measurements were taken from distinct samples or whether the same sample was measured repeatedly |
| ☐ | ☒ | The statistical test(s) used AND whether they are one- or two-sided *Only common tests should be described solely by name; describe more complex techniques in the Methods section.* |
| ☐ | ☒ | A description of all covariates tested |
| ☐ | ☒ | A description of any assumptions or corrections, such as tests of normality and adjustment for multiple comparisons |
| ☐ | ☒ | A full description of the statistical parameters including central tendency (e.g. means) or other basic estimates (e.g. regression coefficient) AND variation (e.g. standard deviation) or associated estimates of uncertainty (e.g. confidence intervals) |
| ☐ | ☒ | For null hypothesis testing, the test statistic (e.g. *F*, *t*, *r*) with confidence intervals, effect sizes, degrees of freedom and *P* value noted *Give P values as exact values whenever suitable.* |
| ☒ | ☐ | For Bayesian analysis, information on the choice of priors and Markov chain Monte Carlo settings |
| ☒ | ☐ | For hierarchical and complex designs, identification of the appropriate level for tests and full reporting of outcomes |
| ☒ | ☐ | Estimates of effect sizes (e.g. Cohen's *d*, Pearson's *r*), indicating how they were calculated |

*Our web collection on statistics for biologists contains articles on many of the points above.*

## Software and code

Policy information about availability of computer code

Data collection | Flow cytometry data were analyzed with Flowjo 10.4 Software. For Metabolite analysis, polar metabolites were quantitatively profiled by a positive/negative ion–switching, targeted liquid chromatography tandem mass spectrometry (LC-MS/MS) based metabolomics platform using a 5500 QTRAP hybrid triple quadruple mass spectrometer (AB/SCIEX) via selected reaction monitoring (SRM). Once the SRM data for ~285 metabolites were acquired, peaks were integrated using a software platform for peak area integration MultiQuant 2.1 (AB/SCIEX). Data analysis was performed using online MetaboAnalyst 3.0 software. Raw sequencing reads were quality-checked using FastQC (v0.11.5) and data were pre-processed with Cutadapt (v2.5) for adapter removal following best practices. Gene expression quantification was performed by aligning against the GRCm38 genome using STAR (v2.7.3aand quantifying reads against Ensembl v98 annotated gene loci with featureCounts (Subread 1.6.2). Differential gene expression analysis was performed using DESeq2 (v1.24.0), while ClusterProfiler (v3.12.0) was utilized for downstream functional investigations. Plots were generated in R using ggplot2 (v3.3.3), EnhancedVolcano (v1.8.0), and ComplexHeatmap (v2.6.2). Storey's q value was utilized to control family-wise error rate. All other statistical analyses were performed using GraphPad Prism (GraphPad Software v.9.4.0)

Data analysis | *Provide a description of all commercial, open source and custom code used to analyse the data in this study, specifying the version used OR state that no software was used.*

For manuscripts utilizing custom algorithms or software that are central to the research but not yet described in published literature, software must be made available to editors and reviewers. We strongly encourage code deposition in a community repository (e.g. GitHub). See the Nature Portfolio guidelines for submitting code & software for further information.

## Data

Policy information about availability of data

All manuscripts must include a data availability statement. This statement should provide the following information, where applicable:
- Accession codes, unique identifiers, or web links for publicly available datasets
- A description of any restrictions on data availability
- For clinical datasets or third party data, please ensure that the statement adheres to our policy

> All data generated or analyzed during this study are included in this published article (and its supplementary information files). Sequencing data have been deposited at the Gene Expression Omnibus database under the accession numbers GSE187394 and GSE206207 and are publicly available.

## Human research participants

Policy information about studies involving human research participants and Sex and Gender in Research.

| | |
|---|---|
| Reporting on sex and gender | N/A |
| Population characteristics | N/A |
| Recruitment | N/A |
| Ethics oversight | N/A |

Note that full information on the approval of the study protocol must also be provided in the manuscript.

# Field-specific reporting

Please select the one below that is the best fit for your research. If you are not sure, read the appropriate sections before making your selection.

☒ Life sciences ☐ Behavioural & social sciences ☐ Ecological, evolutionary & environmental sciences

For a reference copy of the document with all sections, see nature.com/documents/nr-reporting-summary-flat.pdf

# Life sciences study design

All studies must disclose on these points even when the disclosure is negative.

| | |
|---|---|
| Sample size | No statistical methods were used to predetermine sample sizes, and our sample sizes were similar to those reported in previous publications. Strauss, L. et al. Targeted deletion of PD-1 in myeloid cells induces antitumor immunity. Sci Immunol 5, doi:10.1126/sciimmunol.aay1863 (2020). Strauss, L. et al. RORC1 Regulates Tumor-Promoting "Emergency" Granulo-Monocytopoiesis. Cancer cell 28, 253-269, doi:10.1016/j.ccell.2015.07.006 (2015). Molgora, M. et al. TREM2 Modulation Remodels the Tumor Myeloid Landscape Enhancing Anti-PD-1 Immunotherapy. Cell 182, 886-900 e817, doi:10.1016/j.cell.2020.07.013 (2020). |
| Data exclusions | No datapoints were excluded from the analyses. |
| Replication | All experiments were reproducible and were repeated at least three times |
| Randomization | Mice were assigned randomly to the various experimental groups described. Equal numbers of male and female mice were used in all experiments. |
| Blinding | Blinding was not used because the studies involved several genetically engineered mouse strains which are hard to breed in high numbers. For this reason, every available mouse should be used judiciously. |

# Reporting for specific materials, systems and methods

We require information from authors about some types of materials, experimental systems and methods used in many studies. Here, indicate whether each material, system or method listed is relevant to your study. If you are not sure if a list item applies to your research, read the appropriate section before selecting a response.

## Materials & experimental systems

| n/a | Involved in the study |
|-----|----------------------|
| ☐ | ☐ Antibodies |
| ☐ | ☒ Eukaryotic cell lines |
| ☒ | ☐ Palaeontology and archaeology |
| ☐ | ☒ Animals and other organisms |
| ☒ | ☐ Clinical data |
| ☒ | ☐ Dual use research of concern |

## Methods

| n/a | Involved in the study |
|-----|----------------------|
| ☒ | ☐ ChIP-seq |
| ☐ | ☒ Flow cytometry |
| ☒ | ☐ MRI-based neuroimaging |

# Antibodies

| Antibodies used | All antibodies used are described in the methods section and in a supplementary table (Supplementary Table 7). All antibodies were used according to the recommendations of the manufacturer unless indicated otherwise. |
|---|---|
| Validation | *Describe the validation of each primary antibody for the species and application, noting any validation statements on the manufacturer's website, relevant citations, antibody profiles in online databases, or data provided in the manuscript.* |

# Eukaryotic cell lines

Policy information about underline{cell lines and Sex and Gender in Research}

| Cell line source(s) | B16-F10 and MC17-51 cell lines were used and were obtained from ATCC. |
|---|---|
| Authentication | Authentication is provided by the vendor |
| Mycoplasma contamination | All cell lines were tested negative for mycoplasma contamination. In our laboratory, we perform a regular screening for mycoplasma on a monthly basis. |
| Commonly misidentified lines (See ICLAC register) | N/A |

# Animals and other research organisms

Policy information about underline{studies involving animals}; underline{ARRIVE guidelines} recommended for reporting animal research, and underline{Sex and Gender in Research}

| Laboratory animals | Mouse |
|---|---|
| Wild animals | N/A |
| Reporting on sex | Equal number of male and female mice were uses assigned in all experimental groups. This approach was employed because, the study does not involve a disease with distinct prevalence between sex groups. |
| Field-collected samples | N/A |
| Ethics oversight | All mice procedures were approved by the Institutional Animal Care and Use Committee (IACUC) at Beth Israel Deaconess Medical Center (Boston MA), and were in accordance with National Institutes of Health Guidelines for the Care and Use of Animals. |

Note that full information on the approval of the study protocol must also be provided in the manuscript.

# Flow Cytometry

## Plots

Confirm that:

☒ The axis labels state the marker and fluorochrome used (e.g. CD4-FITC).

☒ The axis scales are clearly visible. Include numbers along axes only for bottom left plot of group (a 'group' is an analysis of identical markers).

☒ All plots are contour plots with outliers or pseudocolor plots.

☒ A numerical value for number of cells or percentage (with statistics) is provided.

## Methodology

| Sample preparation | Samples were prepares from live animals. |
|---|---|

| Instrument | Flow cytometry samples were acquired sing Becton Dickinson LSR Fortsessa or Beckman-Coulter Cytoflex For Metabolite analysis, polar metabolites were quantitatively profiled by a positive/negative ion–switching, targeted liquid chromatography tandem mass spectrometry (LC-MS/MS) based metabolomics platform using a 5500 QTRAP hybrid triple quadruple mass spectrometer (AB/SCIEX) via selected reaction monitoring (SRM). Once the SRM data for ~285 metabolites were acquired, peaks were integrated using a software platform for peak area integration MultiQuant 2.1 (AB/SCIEX). Raw sequencing reads were quality-checked using FastQC (v0.11.5) and data were pre-processed with Cutadapt (v2.5) for adapter removal following best practices. Gene expression quantification was performed by aligning against the GRCm38 genome using STAR (v2.7.3aand quantifying reads against Ensembl v98 annotated gene loci with featureCounts (Subread 1.6.2). Differential gene expression analysis was performed using DESeq2 (v1.24.0), while ClusterProfiler (v3.12.0) was utilized for downstream functional investigations. Plots were generated in R using ggplot2 (v3.3.3), EnhancedVolcano (v1.8.0), and ComplexHeatmap (v2.6.2). Storey's q value was utilized to control family-wise error rate. Gene sets used for Gene Set Enrichment. Signals on wester blots were visualized, acquired and quantified with Li-COR Odyssey CLx imaging system. For assessment of cell proliferation 3H-thymidine incorporation was measured using a MicroBeta plate counter was used (TriLux Perkin Elmer). |
|---|---|
| Software | Cells were acquired using Becton Dickinson LSR Fortsessa or Beckman-Coulter Cytoflex Flow cytometer and analyzed with Flowjo Software. |
| Cell population abundance | No cell sorting was used in any of the experiments |
| Gating strategy | All detailed relevant information regarding gatingstrategyis provided in Supplementary figures. |

☒ Tick this box to confirm that a figure exemplifying the gating strategy is provided in the Supplementary Information.

