## [Peer Review File · Nature Immunology]

Peer Review Information

Journal: Nature Immunology

Manuscript Title: SHP-2 and PD-1-SHP-2 signaling regulate myeloid cell differentiation and anti-tumor responses

Corresponding author name(s): Vassiliki Boussiotis

Reviewer Comments & Decisions:

Decision Letter, initial version:

Subject: Decision on Nature Immunology submission NI-A33353

Message: 18th Feb 2022

Dear Dr. Boussiotis,

Your Article, "SHP-2 and canonical PD-1: SHP-2 axis regulate myeloid cell differentiation, anti-tumor responses and innate immune memory" has now been seen by 2 referees. While the work is of potential interest, the reviewers have raised substantial concerns that must be addressed. As such, we cannot accept the current manuscript for publication, but would be interested in considering a revised version that addresses these serious concerns, as long as the novelty of the paper has not been compromised in the interim.

Please revise to thoroughly address all the referees' concerns. We believe it is essential to demonstrate that the observed effects of SHP-2 in TAMs are reflective of the role of PD-1 signaling in these cells (such as investigating the ability of SHP-2 to rescue the phenotype of PD-1-deficient macrophages; comparing the transcriptomic profiles of PD-1-deficient and SHP-2-deficient TAMs or other experimental approaches that could address this issue specifically) and to strengthen the data regarding the interaction between PD-1 and SHP-2 in macrophages in the tumor. If the issues placed by the referees could be addressed, we would invite you to submit a new paper, although we hope you understand that until we have read the revised manuscript in its entirety we cannot promise that it will be sent back for peer review.

At resubmission, please include a "Response to referees" detailing, point-by-point, how you addressed each referee comment. This response will be sent back to the referees along with the revised manuscript.

In addition, please include a revised version of any required reporting checklist. It will be available to referees (and, potentially, statisticians) to aid in their evaluation if the manuscript goes back for peer review. A revised checklist is essential for re-review of the paper. The template Template can be found here:
<https://www.nature.com/documents/nr-reporting-summary.pdf>

You may use the link below to submit your revised manuscript and related files:
[REDACTED]

We hope to receive the revised manuscript within 6 months. If you cannot send it within this time, please let us know. We will be happy to consider your revision so long as nothing similar has been accepted for publication at Nature Immunology or published elsewhere.

Nature Immunology is committed to improving transparency in authorship. As part of our efforts in this direction, we are now requesting that all authors identified as 'corresponding author' on published papers create and link their Open Researcher and Contributor Identifier (ORCID) with their account on the Manuscript Tracking System (MTS), prior to acceptance. ORCID helps the scientific community achieve unambiguous attribution of all scholarly contributions. You can create and link your ORCID from the home page of the MTS by clicking on 'Modify my Springer Nature account'. For more information please visit www.springernature.com/orcid.

Thank you for the opportunity to review your work.

Sincerely,

Ioana Visan, Ph.D.
Senior Editor
Nature Immunology

Tel: 212-726-9207
Fax: 212-696-9752
www.nature.com/ni

Reviewers' Comments:

Reviewer #1:

Remarks to the Author:

In this study, Christofides et al. dissected the PD-1:SH2 axis in myeloid cells in cancer. Through *in vivo* studies in T cell- or myeloid cell-specific SH2-deficient mice, RNA-seq analysis, and adoptive transfer experiments, they convincingly show that myeloid-specific SH2 targeting skewed the myeloid cell fate from immunosuppressive to inflammatory neutrophils and TAMs with features of enhanced antigen-presentation and T cell costimulation capacity, as well as innate anti-tumor immune memory. They further investigated the molecular mechanisms underlying these phenotypes through signaling experiments, and show that the PD-1:SH2 axis inhibits the phosphorylation of two transcription factors, HOXA10 and IRF8, activated by GM-CSF+IL-3 and involved in monocyte lineage commitment.

The study is in line with previous studies of these authors and others, confirming the relevance of PD-1 in myeloid cells and the involvement of SH2 in macrophage activation. In addition to previous publications, this study has the merit to dissect for the first time the molecular mechanisms elucidating the effects of the PD-1:SH2 axis in myeloid differentiation and functional activation in cancer, in response to GM-CSF, thus explaining the tumor resistance of myeloid-specific PD-1-deficient mice.

The study is well conducted, is technically sound and novel. The results presented may have significant impact in new myeloid-based immunotherapy and are of broad interest.

However, I have some suggestions to ameliorate the study:

- In addition to PD-1, SH2 is associated with other receptors in macrophages. For instance, SH2 regulates the IL-10-receptor and STAT3 signaling in human and mouse macrophages and its deficiency protects from colitis and colon cancer (*J Exp Med* (2019) 216 (2): 337–349). Further, in a different pathological context, SH2 regulates TGF- β and MMPs (*J Immunol* 2017; 199:2323-2332). Since both IL-10 and TGF- β are key players in myeloid cell functional activities in cancer, their involvement in the phenotypes described in this study should be investigated. These references should be cited.

-It is unclear whether the phenotype of myeloid cells is sufficient for the anti-tumor activity or the latter depends on T cell activation and recruitment: CD4+ and CD8+ T cell depletion experiments should be performed to clarify this aspect. Indeed, the increased frequency of TEF in the B16-F10 model is not impressive.

- In contrast with the results presented here, it has been proposed that absence of Shp2 in TAMs induces their polarization toward a M2 phenotype through the activation of p-STAT3 and inhibition of p-NF- κ B p65, promoting colorectal-cancer (*Am J Cancer Res.* 2019; 9(9): 1957–1969). These results should be discussed.

-Trained immunity: To better substantiate the adoptive transfer experiment, it would be

important to add the analysis of bone marrow myeloid precursors (GMPs and MDPs) and mature cells in tumor-bearing mice. Figure 6A and B are not necessary.

-Signaling experiments: western blot quantification should be provided, as well as number of experiments and replicates. This part of the study is extremely relevant but not adequately presented.

-The introduction is excessively long: some concepts are also reported in the results and discussion. Conclusions are robust and clearly presented.

Minor points:

-Figure panels are not presented sequentially in the text.

-Graphs show columns and not individual mice. The exact sample size and statistical test are not reported in the legend.

-Line 215-217: the sentence is unclear.

-Ratio of M-MDSC:PMN-MDSC. The graphical representation of the ratio is unclear. Maybe adding the value in the text is better.

-Gating strategy: in the first plot of Fig. S3, the legend Lymph Gate is incorrect. The legend CD11+ in the last plot is wrong (it should be CD11b+). The legend has errors (F4/80+ for M- and G-MDSC).

-Fig S7: the axis legend has not been reported.

Reviewer #4:

Remarks to the Author:

Summary of key results.

The present study describes how Shp2 ablation in myeloid cells results in a solid anti-tumor response by modulating myeloid cell differentiation, function and lineage commitment. In Shp2f/fLysMCre mice, reduction in immunosuppressive MDSCs is paralleled by an increment of activated TEF CD8+ cells and reduced tumor growth, which is not further enhanced by anti-PD-1 treatment. Moreover, growing tumors in Shp2f/fLysMCre were enriched in monocytes and neutrophils with antigen presentation and costimulatory features, suggesting a role of Shp2 in driving myeloid cell commitment. Transfer of tumor educated Shp2f/fLysMCre monocytes, but not neutrophils, into naïve tumor-bearing mice significantly reduced tumor growth, further endorsing the SHP-2 intervention in skewing the emergency myelopoiesis towards an anti-tumor response. Then, the authors provided mechanistic insights, showing that Shp2f/fLysMCre bone marrow cells, in the presence of GM-CSF+IL-13, are enriched in myelocytes, in which persistent phosphorylation HOXA10 and IRF8 promotes an increase in myeloid cells output with monocytic features. This prompted the authors to test whether SHP-2 interacts with PD-1 in monocyte progenitors and indeed, they observed that this interaction occurs and SHP-2 deletion abrogates it.

Originality and significance.

The main issue with this manuscript is the conclusiveness of the data about the PD-1:SHP-2 axis in determining the described alterations. The complex phenotype of lineage-restricted ablation of SHP-2 might depend on the many activities of this phosphatase, regardless of its interaction with PD-1.

There is ample literature about the consequences of myeloid ablation of SHP-2, in many

cases using the very same LysM-Cre as deleter:

- 1) Xu J, Tao B, Guo X, Zhou S, Li Y, Zhang Y, Zhou Z, Cheng H, Zhang X, Ke Y. Macrophage-Restricted Shp2 Tyrosine Phosphatase Acts as a Rheostat for MMP12 through TGF- β Activation in the Prevention of Age-Related Emphysema in Mice. *J Immunol*. 2017 Oct 1;199(7):2323-2332. doi: 10.4049/jimmunol.1601696. Epub 2017 Aug 16. PMID: 28814604.
- 2) Wang S, Yao Y, Li H, Zheng G, Lu S, Chen W. Tumor-associated macrophages (TAMs) depend on Shp2 for their anti-tumor roles in colorectal cancer. *Am J Cancer Res*. 2019 Sep 1;9(9):1957-1969. PMID: 31598397; PMCID: PMC6780667.
- 3) Wen Liu, Ye Yin, Meijing Wang, Ting Fan, Yuyu Zhu, Lihong Shen, Shuang Peng, Jian Gao, Guoliang Deng, Xiangbao Meng, Lingdong Kong, Gen-Sheng Feng, Wenjie Guo, Qiang Xu, Yang Sun, Disrupting phosphatase SHP2 in macrophages protects mice from high-fat diet-induced hepatic steatosis and insulin resistance by elevating IL-18 levels, *Journal of Biological Chemistry*, Volume 295, Issue 31, 2020, Pages 10842-10856, ISSN 0021-9258.
- 4) Guo, W., Liu, W., Chen, Z. et al. Tyrosine phosphatase SHP2 negatively regulates NLRP3 inflammasome activation via ANT1-dependent mitochondrial homeostasis. *Nat Commun* 8, 2168 (2017). <https://doi.org/10.1038/s41467-017-02351-0>.

Major concerns

- 1) The choice of LysMCre as gene deleter might be questionable for the limited and variable penetrance in the myeloid lineage but, in light of the existing literature, a complex phenotype is surely expected. Indeed, Authors do describe in-depth the changes occurring in the myelopoiesis of these mice, also providing molecular downstream candidates (HOXA10 and IRF8). Nonetheless, the link with PD-1 remains feeble.
- 2) In Fig. 7, the coimmunoprecipitation of SHP-2 with PD-1 in bone marrow cells treated with GM-CSF+IL-3 is appealing but expected; it would have been more suggestive to show this interaction in myeloid cells from control tumor-bearing mice.
- 3) In Fig 2, the absence of a statistically significant effect of anti-PD-1 treatment in tumor-bearing Shp2f/fLysMCre mice, as compared to controls, might only be dependent on the mouse numerosity, since there is still some effect. In any case, it is evident that the, which is not equaled by anti-PD-1 therapy (Fig. 2A-B).
- 4) Another example of the complex consequences of SHP-2 ablation is shown in the very last figure. As commented by the same Authors, the effect of PD-L1 interference and SHP-2 ablation are not fully overlapping when HOXA10 and IRF8 phosphorylation are considered.
- 5) There is a personal choice of some markers as surrogates of more complex biology, making the reader ask why they were selected among a panoply of potential candidates: CD44 as a discriminator between activation and exhaustion of CD8+ T lymphocytes in tumor-draining lymph nodes; CD38 as the hallmark of MDSCs (i.e. PMN-MDSCs); the use of splenic and not intra-tumoral myeloid cells to assess the immunosuppressive activity (i.e. the MDSC-like properties). A broader characterization would increase the robustness of the findings.

Minor concerns

- 1) Figure 1D: the proliferation assay is not clear. Generally, a fixed number of effector cells is used versus an increased amount of target cells. In the ratio indicated, are the effectors always 1? If yes, the trend of suppression is not easy to understand. For example, 8 PMN-MDSCs should suppress more (at least in the control mice: Shp2f/f) than 1. Normally the ratio 1:4 (T cell:MDSCs) is supposed to suppress, for a meaningful test. Therefore, are we looking to a very weak suppression? Similar question for Figure 4A.

- 2) In Figure 1G, Y axis label is missing. Same issue for Figure 2F.
- 3) In Supplementary Figure 3, the amount of CD45 positive cells is very low. Is there a compensation problem? The missing FMO for F4/80 is confusing the gating strategy.
- 4) In Supplementary Figure 4 legend, M-MDSCs and PMN-MDSCs are reported to be F4/80+.
- 5) Expression data by RNA-seq are usually validated by either RT-PCR or protein detection.

Author Rebuttal to Initial comments

We would like to thank the editors and the reviewers for the careful evaluation of our manuscript and their thoughtful suggestions for improvement of our studies. In response to the editor's recommendation, we did experiments using mice with PD-1 deficient myeloid cells, performed transcriptomics analysis of TAMs and compared transcriptomics and GSEA signatures of PD-1 deficient TAMs with SHP-2 deficient TAMs. All the comments of the reviewers have also been taken into consideration, additional work has been performed and the manuscript has been revised accordingly. All the changes are highlighted in the revised manuscript.

In response to these recommendations, we performed the following new studies:

1. Generated data of in vivo anti-tumor responses in mice with PD-1 deficient myeloid cells.
2. Examined development of trained immunity in PD-1-deficient monocytes.
3. Performed signaling studies in PD-1-deficient bone marrow myeloid cells.
4. Performed transcriptomics in PD-1-deficient TAMs.
5. Compared transcriptomics and GSEA profiles of PD-1-deficient and SHP-2-deficient TAMs.
6. Performed T cell depletion studies.
7. Assessed bone marrow progenitors.
8. Assessed IRF8 targets in transcriptomics of SHP-2-deficient and PD-1-deficient TAMs.
9. Performed experiments to examine the role of IL-10.
10. Included quantification graphs for all the signaling experiments.
11. Included and discussed additional citations as requested by the referees.

The above studies resulted in the addition of one new multipaned main figure (Figure 8), nine new Supplementary figures (Extended Data Fig. 9, 13, 14, 17-22), three new Supplementary tables (Extended Data Tables, 3, 4, 5) and two new Supplementary GSEA Data Sets (Extended Data Sets 3, 4).

Below is a detailed point-by-point response to the comments of the reviewers.

Reviewer #1:

Remarks to the Author:

In this study, Christofides et al. dissected the PD-1:SH2 axis in myeloid cells in cancer. Through in vivo studies in T cell- or myeloid cell-specific SH2-deficient mice, RNA-seq analysis, and adoptive transfer experiments, they convincingly show that myeloid-specific SH2 targeting skewed the myeloid cell fate from immunosuppressive to inflammatory neutrophils and TAMs with features of enhanced antigen-presentation and T cell costimulation capacity, as well as innate anti-tumor immune memory. They further investigated the molecular mechanisms underlying these phenotypes through signaling experiments, and show that the PD-1:SH2 axis inhibits the phosphorylation of two transcription factors, HOXA10 and IRF8, activated by GM-CSF+IL-3 and involved in monocyte lineage commitment.

The study is in line with previous studies of these authors and others, confirming the relevance of PD-1 in myeloid cells and the involvement of SH2 in macrophage activation. In addition to previous publications, this study has the merit to dissect for the first time the molecular mechanisms elucidating the effects of the PD-1:SH2 axis in myeloid differentiation and functional activation in cancer, in response to GM-CSF, thus explaining the tumor resistance of myeloid-specific PD-1-deficient mice.

The study is well conducted, is technically sound and novel. The results presented may have significant impact in new myeloid-based immunotherapy and are of broad interest.

A: We would like to thank the reviewer for their positive comments regarding the technical quality, novelty and significance of our study.

However, I have some suggestions to ameliorate the study:

- In addition to PD-1, SH2 is associated with other receptors in macrophages. For instance, SH2 regulates the IL-10-receptor and STAT3 signaling in human and mouse macrophages and its deficiency protects from colitis and colon cancer (J Exp Med (2019) 216 (2): 337–349). Further, in a different pathological context, SH2 regulates TGF-beta and MMPs (J Immunol 2017; 199:2323-2332). Since both IL-10 and TGF-beta are key players in myeloid cell functional activities in cancer, their involvement in the phenotypes described in this study should be investigated. These references should be cited.

A: We performed experiments to determine the role of IL-10 in anti-tumor responses in our experimental system. We focused on IL-10 because: a) it was previously reported to be involved in protection from carcinogenesis mediated by SH2 deficiency, as stated by the reviewer (ref#100); b) IL-10 is one of the common transcripts upregulated in SH2-deficient and PD-1deficient TAMs identified by our studies; c) it was recently identified as an IRF8-regulated gene in mature phagocytic cells of the monocytic/DC lineage (ref#94); d) it is currently evolving as a previously unappreciated mediator of anti-tumor responses (ref #98). The results of these new studies are shown in the new Extended Data Figures 21 and 22, and are described in page 22-23 and the relevant discussion section in page 27.

As sue commented on the tentative role of SHP-2 in altering the immunosuppressive properties of myeloid cells by diminishing TGF- β -mediating signaling (page 25) and cited the indicated reference (#107). We did not perform relevant experiments because adding such line of experimentation would be diverting the focus from the main scope of the studies and would make the manuscript unrealistically long.

-It is unclear whether the phenotype of myeloid cells is sufficient for the anti-tumor activity or the latter depends on T cell activation and recruitment: CD4+ and CD8+ T cell depletion experiments should be performed to clarify this aspect. Indeed, the increased frequency of TEF in the B16-F10 model is not impressive.

A: We performed T cell depletion experiments, described in page 14 and shown in the new Extended Data Fig. 14. These studies showed that T cell depletion resulted in a robust increase of tumor growth in $Shp2^{f/f}$ mice but had a much less significant impact in $Shp2^{f/f}LysMCre$ mice, consistent with the conclusion that trained immunity has a major role in the responses of $Shp2^{f/f}LysMCre$ mice.

- In contrast with the results presented here, it has been proposed that absence of Shp2 in TAMs induces their polarization toward a M2 phenotype through the activation of p-STAT3 and inhibition of p-NF- κ B p65, promoting colorectal-cancer (Am J Cancer Res. 2019; 9(9): 1957– 1969). These results should be discussed.

A: We cited this publication (ref #123) and discussed these results in conjunction with other relevant reports from the literature and the findings of our present studies (page 26).

-Trained immunity: To better substantiate the adoptive transfer experiment, it would be important to add the analysis of bone marrow myeloid precursors (GMPs and MDPs) and mature cells in tumor-bearing mice. Figure 6A and B are not necessary.

A: We added the recommended assessments of bone marrow myeloid progenitors and mature cells in mice with SHP-2-deficient myeloid cells in Figure 6. Extended Data Fig. 13 is showing the gating strategy and the current model for myeloid differentiation. We also performed relevant studies with mice with PD-1-deficient myeloid cells (Fig. 8a, b and Extended Data Fig. 18).

-Signaling experiments: western blot quantification should be provided, as well as number of experiments and replicates. This part of the study is extremely relevant but not adequately presented.

A: We provided quantification for all the western blot results (Figure 7) and Extended Data Fig. 16) and added number of experiments and replicates in the figure legend.

-The introduction is excessively long: some concepts are also reported in the results and discussion. Conclusions are robust and clearly presented.

A: We cut and narrowed down the introduction, as suggested.

Minor points:

-Figure panels are not presented sequentially in the text.

A: We made every effort to keep the figure panels sequential inside the figures and present them sequentially in the text. This is quite difficult in the multipaneled figures.

-Graphs show columns and not individual mice. The exact sample size and statistical test are not reported in the legend.

A: We added individual data points in all columns and provided the exact sample size and statistics in the figure legends. Statistical methods are also outlined in a separate section in the Methods.

-Line 215-217: the sentence is unclear.

A: We have completely modified this part of the text.

-Ratio of M-MDSC:PMN-MDSC. The graphical representation of the ratio is unclear. Maybe adding the value in the text is better.

A: We modified the graphical presentation of the M-MDSC: PMN-MDSC ratio to make it more comprehensive (Figure 3h and Extended Figure 4e).

-Gating strategy: in the first plot of Fig. S3, the legend Lymph Gate is incorrect. The legend CD11+ in the last plot is wrong (it should be CD11b+). The legend has errors (F4/80+ for M- and G-MDSC).

A: We apologize for this oversight and we thank the reviewer for catching this error. We corrected the figure legend in Extended Data Fig. 3.

-Fig S7: the axis legend has not been reported.

A: We added the axis legend in this figure.

Reviewer #4:

Remarks to the Author:

Summary of key results.

The present study describes how Shp2 ablation in myeloid cells results in a solid anti-tumor response by modulating myeloid cell differentiation, function and lineage commitment. In Shp2^f/fLysMCre mice, reduction in immunosuppressive MDSCs is paralleled by an increment of activated TEF CD8⁺ cells and reduced tumor growth, which is not further enhanced by anti-PD1 treatment. Moreover, growing tumors in Shp2^f/fLysMCre were enriched in monocytes and neutrophils with antigen presentation and costimulatory features, suggesting a role of Shp2 in driving myeloid cell commitment. Transfer of tumor educated Shp2^f/fLysMCre monocytes, but not neutrophils, into naïve tumor-bearing mice significantly reduced tumor growth, further endorsing the SHP-2 intervention in skewing the emergency myelopoiesis towards an anti-tumor response. Then, the authors provided mechanistic insights, showing that Shp2^f/fLysMCre bone marrow cells, in the presence of GM-CSF+IL-13, are enriched in myelocytes, in which persistent phosphorylation HOXA10 and IRF8 promotes an increase in myeloid cells output with monocytic features. This prompted the authors to test whether SHP-2 interacts with PD-1 in monocyte progenitors and indeed, they observed that this interaction occurs and SHP-2 deletion abrogates it.

Originality and significance.

The main issue with this manuscript is the conclusiveness of the data about the PD-1:SHP-2 axis in determining the described alterations. The complex phenotype of lineage-restricted ablation of SHP-2 might depend on the many activities of this phosphatase, regardless of its interaction with PD-1.

There is ample literature about the consequences of myeloid ablation of SHP-2, in many cases using the very same LysM-Cre as deleter:

1) Xu J, Tao B, Guo X, Zhou S, Li Y, Zhang Y, Zhou Z, Cheng H, Zhang X, Ke Y. Macrophage-Restricted Shp2 Tyrosine Phosphatase Acts as a Rheostat for MMP12 through TGF- β Activation in the Prevention of Age-Related Emphysema in Mice. *J Immunol.* 2017 Oct 1;199(7):23232332. doi: 10.4049/jimmunol.1601696. Epub 2017 Aug 16. PMID: 28814604.

- 2) Wang S, Yao Y, Li H, Zheng G, Lu S, Chen W. Tumor-associated macrophages (TAMs) depend on Shp2 for their anti-tumor roles in colorectal cancer. *Am J Cancer Res.* 2019 Sep 1;9(9):1957-1969. PMID: 31598397; PMCID: PMC6780667.
- 3) Wen Liu, Ye Yin, Meijing Wang, Ting Fan, Yuyu Zhu, Lihong Shen, Shuang Peng, Jian Gao, Guoliang Deng, Xiangbao Meng, Lingdong Kong, Gen-Sheng Feng, Wenjie Guo, Qiang Xu, Yang Sun, Disrupting phosphatase SHP2 in macrophages protects mice from high-fat diet-induced hepatic steatosis and insulin resistance by elevating IL-18 levels, *Journal of Biological Chemistry*, Volume 295, Issue 31, 2020, Pages 10842-10856, ISSN 0021-9258.
- 4) Guo, W., Liu, W., Chen, Z. et al. Tyrosine phosphatase SHP2 negatively regulates NLRP3 inflammasome activation via ANT1-dependent mitochondrial homeostasis. *Nat Commun* 8, 2168 (2017). <https://doi.org/10.1038/s41467-017-02351-0>.

A: We agree with the reviewer that the effects of SHP-2 are mediated by multiple interactors and not simply by PD-1. We performed relevant analysis and discussed this in the manuscript. However, there is a significant overlap in the molecular and signaling implications induced by deletion of PD-1 with those induced by deletion of SHP-2, as determined by transcriptomics and GSEA studies. We also agree that there is ample literature related to the consequences of myeloid ablation of SHP-2 using the very same *LysM-Cre* as deleter. This is because this approach is the only technically feasible means to achieve deletion with SHP-2 in myeloid cells resulting in a viable mouse. Other approaches of hematopoietic or global SHP-2 deletion are lethal. However, each of these studies addressed a different question or investigated the role of SHP-2 in a different context. This ample literature indicates the interest in this very subject in the field of oncology and tumor immunology.

Major concerns

- 1) The choice of *LysM-Cre* as gene deleter might be questionable for the limited and variable penetrance in the myeloid lineage but, in light of the existing literature, a complex phenotype is surely expected. Indeed, Authors do describe in-depth the changes occurring in the myelopoiesis of these mice, also providing molecular downstream candidates (*HOXA10* and *IRF8*). Nonetheless, the link with PD-1 remains feeble.

A: In the revised manuscript, we added new experiments performed with PD-1 deficient myelocytes (Fig. 7u-x). In these cells the effect of the PD-1 pathway is more clearly revealed than by the use of the anti-PD-L1 antibody. Yet, the pattern of enhanced phosphorylation of *HOXA10* and *IRF8* in both approaches is similar (Fig. 7u-x and Supplementary Fig. 16). Please, also see response to point (4) below.

- 2) In Fig. 7, the coimmunoprecipitation of SHP-2 with PD-1 in bone marrow cells treated with GM-CSF+IL-3 is appealing but expected; it would have been more suggestive to show this interaction in myeloid cells from control tumor-bearing mice.

A: We respectfully disagree that the coimmunoprecipitation of SHP-2 with PD-1 in bone marrow cells treated with GM-CSF+IL3 is expected. PD-1: SHP-2 interaction has been extensively studied in lymphocytes. However, canonical PD-1: SHP2 interaction in myeloid cells in response to a major growth factor that drives myeloid cell expansion, activation and differentiation, has never been identified. Unfortunately, detection of this interaction in myeloid cells from a tumor-bearing mouse is not technically feasible. Interaction of PD-1 with SHP-2 depends on PD-1 phosphorylation, a protein modification that cannot be preserved in material collected from an animal, which requires harvesting, prolonged tissue processing and magnetic isolation of myeloid cells, conditions too harsh to preserve protein phosphorylation. For the same reason, all the published studies regarding PD-1:SHP-2 interaction in T cells have been performed using in vitro approaches, frequently by overexpressing the proteins of interest in transfected cells. In our studies, we were able to detect, for the first time, interaction of endogenous PD-1 with SHP-2 in myeloid cells in response to GM-CSF, a growth factor regulating normal and emergency myelopoiesis (Fig. 7p-t).

3) In Fig 2, the absence of a statistically significant effect of anti-PD-1 treatment in tumor-bearing Shp2f/fLysMCre mice, as compared to controls, might only be dependent on the mouse numerosity, since there is still some effect. In any case, it is evident that the, which is not equaled by anti-PD-1 therapy (Fig. 2A-B).

A: We agree that there is some effect of the PD-1 blocking therapy in the Shp2f/fLysMCre mice. However, this treatment is significantly less impactful in Shp2f/fLysMCre mice compared to control tumor-bearing mice. This conclusion is based on data generated by standard statistical analysis.

4) Another example of the complex consequences of SHP-2 ablation is shown in the very last figure. As commented by the same Authors, the effect of PD-L1 interference and SHP-2 ablation are not fully overlapping when HOXA10 and IRF8 phosphorylation are considered.

A: We agree that the effect of anti-PD-L blocking antibody and SHP-2 ablation are not fully overlapping when HOXA10 and IRF8 phosphorylation are considered. We have included this clarification in the manuscript (page 17). However, we also did new experiments with PD-1 deficient myelocytes. In these cells, the effect of the PD-1 pathway is more clearly revealed than by the use of the anti-PD-L1 antibody. These results showed a significant impact on phosphorylation of HOXA10 and IRF8 (Fig. 7u-x). Thus, PD-1-deficiency in myelocytes and antibody-mediated blockade of the pathway induced similar patterns of altered phosphorylation but of different magnitude (Fig. 7u-x and Supplementary Fig. 16).

5) There is a personal choice of some markers as surrogates of more complex biology, making the reader ask why they were selected among a panoply of potential candidates: CD44 as a discriminator between activation and exhaustion of CD8+ T lymphocytes in tumor-draining lymph nodes; CD38 as the

hallmark of MDSCs (i.e. PMN-MDSCs); the use of splenic and not intra-tumoral myeloid cells to assess the immunosuppressive activity (i.e. the MDSC-like properties). A broader characterization would increase the robustness of the findings.

A: We would like to clarify that we are not using CD44 as a discriminator between activation and exhaustion. CD44 is a marker expressed in various activated T cell subsets such as effector, effector memory and central memory and this is the reason for which we used it in our studies. To further clarify that our purpose is to assess activation/effector status, we added results of Ifn-g production in the different experimental conditions (Fig. 1d). We have assessed exhaustion markers in a separate experiment (Extended Data Fig. 5).

CD38 has been recently proposed as an informative marker of MDSC (ref#35). Unfortunately, there are no unique surface markers identifying this immunosuppressive myeloid population. For this reason, the most reliable characterization of MDSC is the assessment of their immunosuppressive function in a well-established suppression assay of antigen-specific responses using splenocytes from OTI mice. In our studies we have used this approach to characterize the functional properties of MDSC (Fig.1e, Fig. 4a, Extended Data Fig. 7, Extended Data Fig. 21g, h). Splenic MDSC in the suppression assay are widely used in the literature because the number of MDSC that can be isolated from the tumor are usually insufficient for this purpose. Splenic MDSC and tumor MDSC represent the same cell population originating from the bone marrow.

Minor concerns

1) Figure 1D: the proliferation assay is not clear. Generally, a fixed number of effector cells is used versus an increased amount of target cells. In the ratio indicated, are the effectors always 1? If yes, the trend of suppression is not easy to understand. For example, 8 PMN-MDSCs should suppress more (at least in the control mice: Shp2f/f) than 1. Normally the ratio 1:4 (T cell:MDSCs) is supposed to suppress, for a meaningful test. Therefore, are we looking to a very weak suppression? Similar question for Figure 4A.

A: We provided a detailed description of the standard suppression assay in the Method section. Clear indication of the cell ratios was added in the relevant figures (Fig. 1, Fig. 4, Extended Data Fig. 7 and Extended Data Fig. 21).

2) In Figure 1G, Y axis label is missing. Same issue for Figure 2F.

A: We added the missing labels in these panels. Apologies for this oversight and thank you for noticing it.

3) In Supplementary Figure 3, the amount of CD45 positive cells is very low. Is there a compensation problem? The missing FMO for F4/80 is confusing the gating strategy.

A: The fraction of CD45-positive cells in this gating strategy example is low because there is a high fraction of CD45-negative tumor cells.

4) In Supplementary Figure 4 legend, M-MDSCs and PMN-MDSCs are reported to be F4/80+.

A: We corrected the figure legend of Extended Data Fig. 3. We apologize for the oversight and we thank the reviewer for noticing this error.

5) Expression data by RNA-seq are usually validated by either RT-PCR or protein detection.

A: We agree that this used to be a standard practice in the past but as now RNA sequencing technologies have been optimized and RNAseq studies have been streamlined, this is no longer a standard approach in most publications. We have assessed protein by flow cytometry for several of the differentially expressed genes identified by RNAseq including CD86, Irf8, Ifng, IL-10 (Fig. 5m-r). As the reviewer suggested we also performed qPCR for two representative genes in SHP-2 deficient and control TAMs (Extended Data Fig. 9).

Decision Letter, first revision:

Subject: Your manuscript, NI-A33353A

Message: Our ref: NI-A33353A

17th Oct 2022

Dear Dr. Boussiotis,

Thank you for your patience as we've prepared the guidelines for final submission of your Nature Immunology manuscript, "SHP-2 and canonical PD-1: SHP-2 axis regulate myeloid cell differentiation, anti-tumor responses and innate immune memory" (NI-A33353A). Please carefully follow the step-by-step instructions provided in the attached file, and add a response in each row of the table to indicate the changes that you have made. Please also check and comment on any additional marked-up edits we have proposed within the text. Ensuring that each point is addressed will help to ensure that your revised manuscript can be swiftly handed over to our production team.

We would like to start working on your revised paper, with all of the requested files and forms, as soon as possible (preferably within one week). Please get in contact with us if you anticipate delays.

When you upload your final materials, please include a point-by-point response to any remaining reviewer comments and please make sure to upload your checklist.

If you have not done so already, please alert us to any related manuscripts from your group that are under consideration or in press at other journals, or are being written up for submission to other journals (see: <https://www.nature.com/nature-portfolio/editorial-policies/plagiarism#policy-on-duplicate-publication> for details).

In recognition of the time and expertise our reviewers provide to Nature Immunology's editorial process, we would like to formally acknowledge their contribution to the external peer review of your manuscript entitled "SHP-2 and canonical PD-1: SHP-2 axis regulate myeloid cell differentiation, anti-tumor responses and innate immune memory". For those reviewers who give their assent, we will be publishing their names alongside the published article.

Nature Immunology offers a Transparent Peer Review option for new original research manuscripts submitted after December 1st, 2019. As part of this initiative, we encourage our authors to support increased transparency into the peer review process by agreeing to have the reviewer comments, author rebuttal letters, and editorial decision letters published as a Supplementary item. When you submit your final files please clearly state in your cover letter whether or not you would like to participate in this initiative. Please note that failure to state your preference will result in delays in accepting your manuscript for publication.

Cover suggestions

As you prepare your final files we encourage you to consider whether you have any images or illustrations that may be appropriate for use on the cover of Nature Immunology.

Nature Immunology has now transitioned to a unified Rights Collection system which will allow our Author Services team to quickly and easily collect the rights and permissions required to publish your work. Approximately 10 days after your paper is formally accepted, you will receive an email in providing you with a link to complete the grant of rights. If your paper is eligible for Open Access, our Author Services team will also be in touch regarding any additional information that may be required to arrange payment for your article.

Please note that *Nature Immunology* is a Transformative Journal (TJ). Authors may publish their research with us through the traditional subscription access route or make their paper immediately open access through payment of an article-processing charge (APC). Authors will not be required to make a final decision about access to their article until it has been accepted. [Find out more about Transformative Journals](https://www.springernature.com/gp/open-research/transformative-journals).

If you have any questions about costs, Open Access requirements, or our legal forms, please contact ASJournals@springernature.com.

Please use the following link for uploading these materials: [REDACTED]

Best regards,

Elle Morris
Senior Editorial Assistant
Nature Immunology
Phone: 212 726 9207
Fax: 212 696 9752
E-mail: immunology@us.nature.com

On behalf of

Ioana Visan, Ph.D.
Senior Editor
Nature Immunology

Tel: 212-726-9207
Fax: 212-696-9752
www.nature.com/ni

Reviewer #1:

Remarks to the Author:

The authors addressed my questions.

However, there are few sentences to be revised. For instance:

In the abstract: PD-1 checkpoint inhibitor induces T cell inactivation... It should be: The PD-1 checkpoint induces T cell inactivation...

Line 130:....were present in bone marrow.

Line 296: The differentially.... The sentence is not correct.

Reviewer #4:

Remarks to the Author:

No additional comments.

Final Decision Letter:

Subject: Decision on Nature Immunology submission NI-A33353B

Message: In reply please quote: NI-A33353B

Dear Dr. Boussiotis,

I am delighted to accept your manuscript entitled "SHP-2 and PD-1-SHP-2 signaling regulate myeloid cell differentiation and anti-tumor responses" for publication in an upcoming issue of Nature Immunology.

Over the next few weeks, your paper will be copyedited to ensure that it conforms to Nature Immunology style. Once your paper is typeset, you will receive an email with a link to choose the appropriate publishing options for your paper and our Author Services team will be in touch regarding any additional information that may be required.

Please note that *Nature Immunology* is a Transformative Journal (TJ). Authors may publish their research with us through the traditional subscription access route or make their paper immediately open access through payment of an article-processing charge (APC). Authors will not be required to make a final decision about access to their article until it has been accepted. [Find out more about Transformative Journals](https://www.springernature.com/gp/open-research/transformative-journals).

Authors may need to take specific actions to achieve [compliance with funder and institutional open access mandates](https://www.springernature.com/gp/open-research/funding/policy-compliance-faqs). If your research is supported by a funder that requires immediate open access (e.g. according to [Plan S principles](https://www.springernature.com/gp/open-research/plan-s-compliance)) then you should select the gold OA route, and we will direct you to the compliant route where possible. For authors selecting the subscription publication route, the journal's standard licensing terms will need to be accepted, including [self-archiving policies](https://www.springernature.com/gp/open-research/policies/journal-policies). Those licensing terms will supersede any other terms that the author or any third party may assert apply to any version of the manuscript.

Your paper will be published online soon after we receive your corrections and will appear in print in the next available issue. Content is published online weekly on Mondays and Thursdays, and the embargo is set at 16:00 London time (GMT)/11:00 am US Eastern time (EST) on the day of publication. Now is the time to inform your Public Relations or Press Office about your paper, as they might be interested in promoting its publication. This will allow them time to prepare an accurate and satisfactory press release. Include your manuscript tracking number (NI-A33353B) and the name of the journal, which they will need when they contact our office.

About one week before your paper is published online, we shall be distributing a press release to news organizations worldwide, which may very well include details of your work. We are happy for your institution or funding agency to prepare its own press release, but it must mention the embargo date and *Nature Immunology*. Our Press Office will contact you closer to the time of publication, but if you or your Press Office have any enquiries in the meantime, please contact press@nature.com.

Also, if you have any spectacular or outstanding figures or graphics associated with your manuscript - though not necessarily included with your submission - we'd be delighted to consider them as candidates for our cover. Simply send an electronic version (accompanied by a hard copy) to us with a possible cover caption enclosed.

If you have not already done so, we strongly recommend that you upload the step-by-step protocols used in this manuscript to the Protocol Exchange. Protocol Exchange is an open online resource that allows researchers to share their detailed experimental know-how. All uploaded protocols are made freely available, assigned DOIs for ease of citation and fully searchable through nature.com. Protocols can be linked to any publications in which they are used and will be linked to from your article. You can also establish a dedicated page to collect all your lab Protocols. By uploading your Protocols to Protocol Exchange, you are enabling researchers to more readily reproduce or adapt the methodology you use, as well as increasing the visibility of your protocols and papers. Upload your Protocols at www.nature.com/protocolexchange/. Further information can be found at www.nature.com/protocolexchange/about .

Please note that we encourage the authors to self-archive their manuscript (the accepted version before copy editing) in their institutional repository, and in their funders' archives, six months after publication. Nature Portfolio recognizes the efforts of funding bodies to increase access of the research they fund, and strongly encourages authors to participate in such efforts. For information about our editorial policy, including license agreement and author copyright, please visit www.nature.com/ni/about/ed_policies/index.html

Sincerely,

Ioana Visan, Ph.D.
Senior Editor
Nature Immunology

Tel: 212-726-9207
Fax: 212-696-9752
www.nature.com/ni